# Mouse T cell priming is enhanced by maturation-dependent stiffening of the dendritic cell cortex

**Daniel Blumenthal\*, Vidhi Chandra, Lyndsay Avery, Janis K Burkhardt\***

Department of Pathology and Laboratory Medicine, Children's Hospital of Philadelphia Research Institute and Perelman School of Medicine at the University of Pennsylvania, Philadelphia, United States

**Abstract** T cell activation by dendritic cells (DCs) involves forces exerted by the T cell actin cytoskeleton, which are opposed by the cortical cytoskeleton of the interacting antigen-presenting cell. During an immune response, DCs undergo a maturation process that optimizes their ability to efficiently prime naïve T cells. Using atomic force microscopy, we find that during maturation, DC cortical stiffness increases via a process that involves actin polymerization. Using stimulatory hydrogels and DCs expressing mutant cytoskeletal proteins, we find that increasing stiffness lowers the agonist dose needed for T cell activation. CD4$^+$ T cells exhibit much more profound stiffness dependency than CD8$^+$ T cells. Finally, stiffness responses are most robust when T cells are stimulated with pMHC rather than anti-CD3ε, consistent with a mechanosensing mechanism involving receptor deformation. Taken together, our data reveal that maturation-associated cytoskeletal changes alter the biophysical properties of DCs, providing mechanical cues that costimulate T cell activation.

**\*For correspondence:**
daniel.blumenthal.chop@gmail.com (DB);
jburkhar@pennmedicine.upenn.edu (JKB)

**Competing interests:** The authors declare that no competing interests exist.

## Introduction

The initiation of an adaptive immune response requires priming of naïve T cells by professional antigen-presenting cells (APCs). This process involves multiple receptor-ligand interactions, which occur in concert at a specialized cell-cell contact site called the immunological synapse (*Dustin, 2014*). Through these interactions, APCs transmit a highly orchestrated series of signals that induce T cell activation and direct differentiation of T cell populations (*Friedl et al., 2005*). While the biochemical aspects of this process have been the subject of many studies, the contribution of mechanical cues is only now being uncovered.

Following initial T cell receptor (TCR) engagement, T cells apply cycles of pushing and pulling forces on interacting APCs (*Bashour et al., 2014*; *Blumenthal and Burkhardt, 2020*; *Hui et al., 2015*; *Husson et al., 2011*; *Sawicka et al., 2017*). These forces are essential for proper T cell activation (*Li et al., 2010*; *Pryshchep et al., 2014*). Moreover, force application is responsible, at least in part, for the ability of T cells to rapidly discriminate between agonist and antagonist antigens (*Das et al., 2015*; *Liu et al., 2014*). While the mechanism by which force is translated into biochemical cues remains controversial (*Das et al., 2015*; *Hong et al., 2015*; *Kim et al., 2009*), there is evidence that early tyrosine phosphorylation events downstream of TCR engagement occur at sites where applied force is maximal (*Bashour et al., 2014*). Interestingly, the amount of force a T cell applies is directly affected by the stiffness of the stimulatory substrate (*Husson et al., 2011*; *Sawicka et al., 2017*). Thus, it appears that force application is mechanically coupled to the T cell's ability to sense stiffness (mechanosensing). In other cell types, substrate stiffness has been shown to affect a variety of cell functions including differentiation, migration, growth, and survival (*Byfield et al., 2009*; *Discher et al., 2005*; *Engler et al., 2006*; *Lo et al., 2000*; *Oakes et al., 2009*;

*Pelham and Wang, 1997*; *Solon et al., 2007*; *Trappmann et al., 2012*). Stiffness sensing in T cells has not been well studied, though there is some evidence that substrate stiffness affects both initial priming and effector functions (*Alatoom et al., 2020*; *Basu et al., 2016*; *Judokusumo et al., 2012*; *O'Connor et al., 2012*; *Saitakis et al., 2017*). Since the physiologically relevant substrate for T cell priming is the surface of the interacting APC, one might predict that changes in cortical stiffness of the APC will profoundly influence T cell priming. However, this prediction remains untested, and studies addressing the role of substrate stiffness in T cell priming did not take into consideration the physiological stiffness of APCs.

Dendritic cells (DCs) are the dominant APCs that prime T cells in vivo (*Jung et al., 2002*). One of the hallmarks of DC biology is the process of maturation. Immature DCs are sentinels of the immune system, specialized for immune surveillance and antigen processing (*Mellman and Steinman, 2001*). In response to infection or injury, inflammatory stimuli trigger signaling pathways that induce molecular reprogramming of the cell. The resulting mature DCs express high levels of surface ligands and cytokines needed for efficient T cell priming (*Burns et al., 2004*). A central feature of DC maturation is remodeling of the actin cytoskeleton, a process that underlies other maturation-associated changes such as downregulation of endocytosis and increased migratory behavior (*Garrett et al., 2000*; *West et al., 2000*). Cytoskeletal remodeling also has a direct impact on the ability of mature DCs to prime T cells (*Al-Alwan et al., 2001*; *Comrie et al., 2015*). Indeed, depolymerization of actin filaments perturbs the ability of mature peptide-pulsed DCs to activate T cells, indicating that actin plays an important role on the DC side of the immunological synapse. We hypothesized that maturation-associated changes in the actin cytoskeleton modulate the stiffness of the DC cortex and promote T cell priming by providing physical resistance to the pushing and pulling forces exerted by the interacting T cell.

In this study, we aimed to better understand the relationship between DC cortical stiffness and T cell activation. We found that during maturation, DCs undergo a 2–3-fold increase in cortical stiffness and that T cell activation is sensitive to stiffness over the same range. Stiffness sensitivity was observed in all T cell populations tested and was particularly robust in naïve CD4$^+$ T cells. Moreover, stiffness responses were most profound when T cells were engaged through TCR$\alpha\beta$ directly, consistent with a mechanosensing mechanism involving receptor deformation. Since we find that stiffer surfaces lower the threshold signal required for T cell activation, we conclude that stiffness serves as a novel biophysical costimulatory mechanism that functions in concert with canonical signaling cues to facilitate T cell priming.

## Results

### Dendritic cell stiffness increases upon maturation

During maturation, DCs undergo a set of phenotypic changes that transform them into highly effective APCs (*Mellman and Steinman, 2001*). We hypothesized that as part of this maturation process, DCs might also modulate their cortical stiffness. To test this, we used atomic force microscopy (AFM) to directly measure cortical stiffness of immature and mature DCs. Murine bone marrow-derived DCs (BMDCs) were prepared as described in the Materials and methods and cultured in the absence or presence of lipopolysaccharide to induce maturation. Cells were plated on poly L-lysine (PLL) coated coverslips and allowed to spread for at least 4 hr before the measurement of cortical stiffness by AFM microindentation. Because the population of LPS-treated cells was heterogeneous with respect to maturation markers, cells were labeled with fluorescent anti-CD86, and immature (CD86 negative) or mature (CD86 high) cells were selected for AFM measurements (*Figure 1—figure supplement 1*). As shown in *Figure 1A*, immature BMDCs were quite soft, with a mean Young's modulus of 2.2 ± 1.7 kPa. Mature BMDCs were almost 2-fold stiffer, with a Young's modulus of 3.3 ± 1.4 kPa. Importantly, the stiffness of CD86-negative BMDCs within the LPS-treated population was 2.0 ± 1.0, the same as that of untreated, immature DCs. This demonstrates that the observed increase in stiffness is a property of DC maturation rather than an unrelated response to lipopolysaccharide treatment. Since BMDCs do not recapitulate all of the properties of classical, tissue-resident DCs (*Guilliams and Malissen, 2015*; *Lutz et al., 2016*; *Na et al., 2016*), we verified our results by measuring the stiffness of ex vivo DCs purified from spleens of untreated or LPS-injected mice. Results were very similar; the stiffness of immature splenic DCs was 1.9 ± 0.7, and lipopolysaccharide

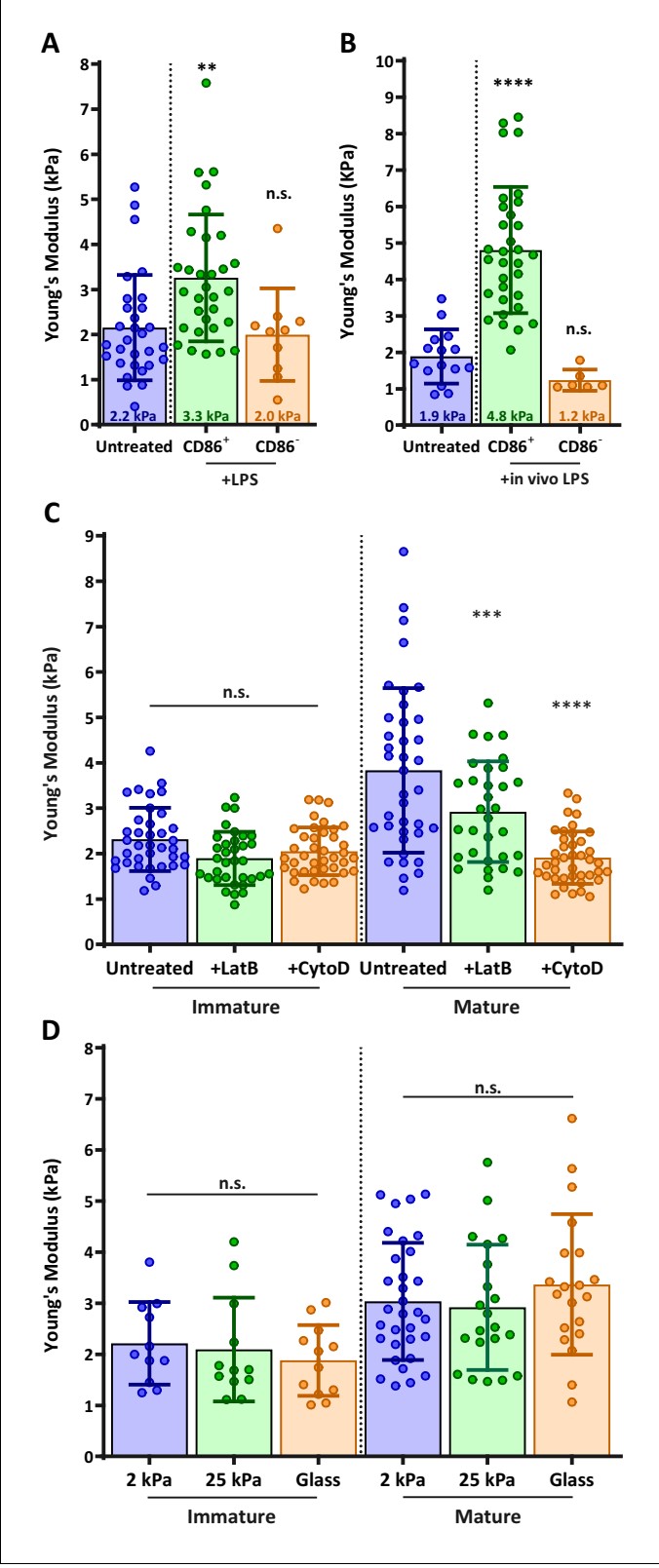

**Figure 1.** DC maturation induces an actin-dependent increase in cortical stiffness. (**A**) BMDCs were untreated or matured by treatment with LPS, and cortical stiffness was measured by AFM microindentation. Fluorescent anti-CD86 labeling was used to select immature (CD86-negative) or mature (CD86 high) cells. (**B**) Ex vivo DCs were purified from spleens of untreated mice or from mice injected with lipopolysaccharide 24 hr before harvesting the

*Figure 1 continued on next page*

*Figure 1 continued*

spleen, and analyzed as in A. (**C**) Immature or LPS-matured BMDCs either left untreated or treated with 10 μm of the actin depolymerizing agents latrunculin-B or cytochalasin-D before AFM measurements. (**D**) Immature or LPS-matured BMDCs were plated on substrates of different stiffness before AFM measurements. Data are pooled from 2 to 3 independent experiments. Each data point represents an average of two stiffness measurements at different locations around a single cell nucleus. Error bars denote standard deviation. n.s non-significant, **p<0.01, ***p<0.001, ****p<0.0001 calculated by an unpaired one-way ANOVA, post-hoc Tukey corrected test. The online version of this article includes the following source data and figure supplement(s) for figure 1:

**Source data 1.** Figure 1 - data table.
**Figure supplement 1.** Selecting mature or immature BMDCs based on CD86 staining.
**Figure supplement 2.** Analysis of BMDC spreading on hydrogel and glass surfaces.
**Figure supplement 2—source data 1.** Figure 1 - figure supplement 2 - data table.
**Figure supplement 3.** Validation of hydrogel compliances.
**Figure supplement 3—source data 1.** Figure 1 - figure supplement 3 - data table.
**Figure supplement 4.** Indentation length and Hertzian model fitting on AFM data.
**Figure supplement 4—source data 1.** Figure 1 - figure supplement 4 - data table.

treatment induced a 2.5-fold increase in stiffness resulting in mature splenic DCs with a mean Young's modulus of 4.8 ± 1.7 kPa (*Figure 1B*). These results demonstrate that stiffness modulation is a bona fide trait of DC maturation. Moreover, they establish that the biologically relevant range of DC stiffness lies between 2 and 8 kPa.

## The maturation-induced increase in stiffness is actin dependent and substrate independent

One well-known feature of DC maturation is remodeling of the actin cytoskeleton. This process involves changes in the activation status of Rho GTPases and downstream actin regulatory proteins, and it is known to downregulate antigen uptake and increase cell motility (*Garrett et al., 2000*; *West et al., 2000*). To know if changes in actin cytoarchitecture can also result in increased cortical stiffness, we treated immature and mature BMDCs with actin-depolymerizing agents cytochalasin-D or latrunculin-B. Neither drug affected the stiffness of immature BMDCs, indicating that the basal level of stiffness depends on factors other than the actin cytoskeleton (*Figure 1C*). By contrast, both drugs induced a significant decrease in the stiffness of mature DCs, with cytochalasin reducing their stiffness to that of immature DCs. We conclude that the increased cortical stiffness observed upon DC maturation is another feature of actin cytoskeletal reprogramming.

Some cell types regulate their stiffness in response to the stiffness of their substrate (*Byfield et al., 2009*; *Tee et al., 2011*). To test whether DCs exhibit this behavior, immature and mature BMDCs were plated on PLL-coated substrates of different compliances (hydrogels of 2 or 25 kPa, or glass surfaces in the GPa range) and allowed to spread on the surface for at least 4 hr before AFM measurement. Importantly, when allowed to spread on different PLL-coated surfaces, no notable changes were detected in BMDC morphology, and changes in cell spreading area were minimal (less than 10%, *Figure 1—figure supplement 2*). Hydrogel compliance was verified by measuring the elastic modulus of the surface in areas devoid of cells (*Figure 1—figure supplement 3*). As shown in *Figure 1D*, substrate compliance did not affect cortical stiffness of either immature or mature BMDCs. In control studies, we could readily detect substrate-dependent changes in stiffness of normal fibroblast cells (not shown). Thus, we conclude that DCs maintain a specific cortical stiffness, which is characteristic of their maturation state.

## DC cortical stiffness is primarily controlled by actin polymerization

Next, we sought to identify the molecular mechanisms controlling DC cortical stiffness. Several actin regulatory mechanisms are known to change during DC maturation. In particular, mature DCs upregulate the actin bundling protein fascin (*Yamashiro, 2012*), show activation of myosin-dependent processes (*van Helden et al., 2008*), and undergo changes in the activation and localization of Rho-GTPases, which in turn regulate actin polymerization via the Arp2/3 complex and formins (*Burns et al., 2001*; *Garrett et al., 2000*; *West et al., 2000*). To ask how each of these pathways influences cortical stiffness, we used small molecule inhibitors and DCs from relevant knockout mice.

Note that to facilitate comparison between experiments, control immature and mature DCs were tested in each experiment, and results were normalized based on values for mature DCs. First, we tested the role of fascin, which is known to generate very stiff actin bundles in vitro (*Démoulin et al., 2014*). Surprisingly, the stiffness of BMDCs from fascin[-/-] mice was not significantly different from that of WT BMDCs either before or after LPS-induced maturation (*Figure 2A*). Next, we tested the contribution of myosin contractility, which is known to control stiffness and membrane tension in other cell types (*Salbreux et al., 2012*). As shown in *Figure 2B*, treating mature BMDCs with the myosin II inhibitor blebbistatin reduced stiffness by a small, albeit statistically significant amount. Similar results were obtained with the Rho-kinase (ROCK) inhibitor Y27623, which indirectly inhibits myosin function.

Next, we considered the possibility that cortical stiffness is modulated by actin polymerization. Broadly speaking, actin polymerization is induced by two sets of proteins: formins generate linear actin filaments, while activators of the Arp2/3 complex produce branched actin structures. Treatment of DCs with the pan-formin inhibitor SMIFH2 significantly reduced the cortical stiffness of mature DCs (*Figure 2B*). The most profound reduction was observed after inhibition of Arp2/3-mediated branched actin polymerization by CK666. DCs express multiple activators of Arp2/3 complex, of which two have been implicated in maturation-associated changes in actin architecture:

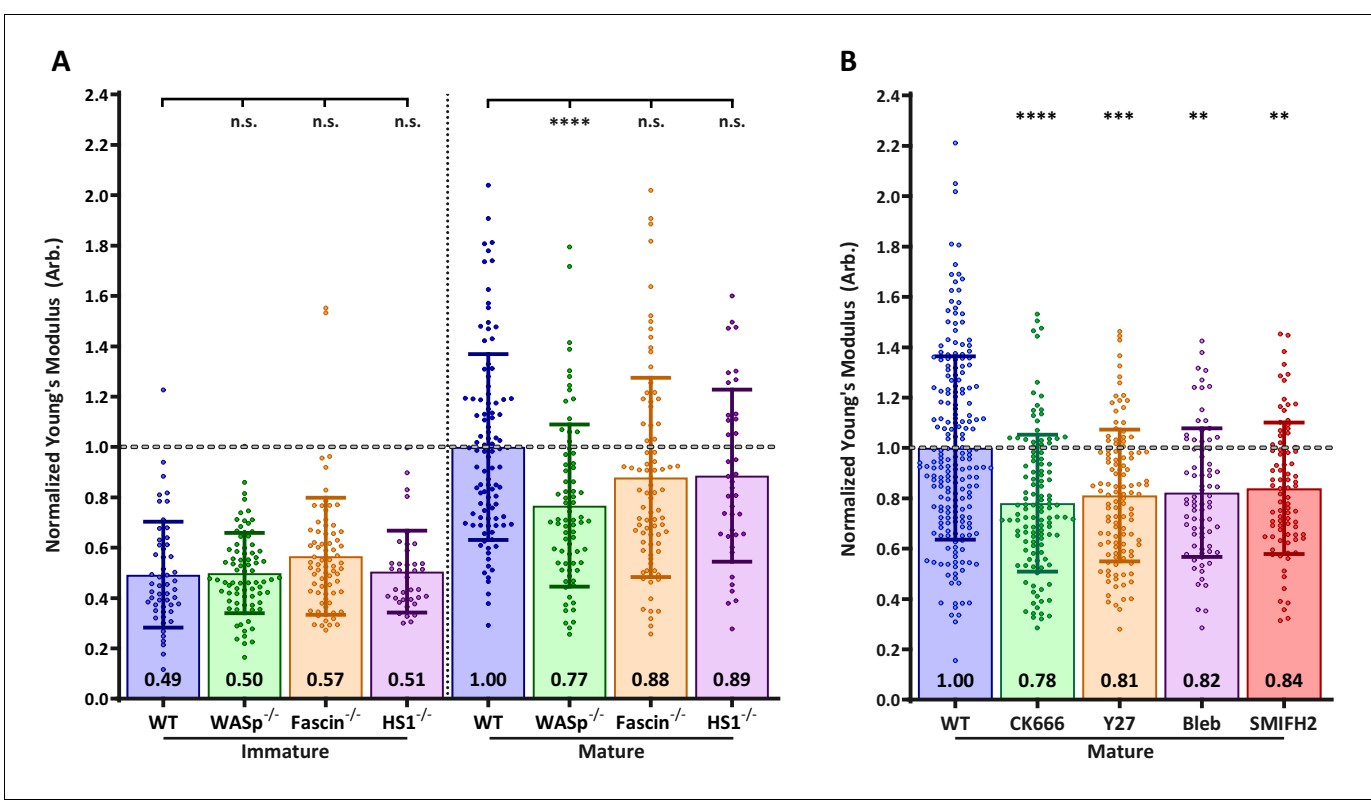

**Figure 2.** Effects of actin regulatory proteins on DC cortical stiffness. Murine BMDCs from WT mice or mice lacking key actin-associated proteins were untreated or matured by treatment with LPS, and cortical stiffness was measured by AFM microindentation. To facilitate comparison between experiments, results were normalized to values of mature WT BMDCs in each experiment. (**A**) Cortical stiffness of BMDCs from mice lacking important actin modulating proteins. (**B**) Cortical stiffness of WT BMDCs treated with cytoskeletal inhibitors. CK666 (100 µM) was used to inhibit branched actin polymerization by Arp2/3. SMIFH2 (10 µm) was used to inhibit linear actin polymerization by formins. Acto-myosin contractility was inhibited directly with blebbstatin (50 µM) or indirectly with the Rho-kinase inhibitor Y27632 (25 µM). All drugs do not affect immature BMDCs (data not shown). Data points for untreated WT BMDCs (both immature and mature) were pooled from all experiments as a reference. The dashed line represents the average stiffness of untreated mature WT BMDCs from all experiments. Data are pooled from two to three independent experiments. Each data point represents an average of two stiffness measurements at different locations around a single cell nucleus. Error bars denote standard deviation. n.s non-significant, *p<0.05, **p<0.01, ***p<0.001, ****p<0.0001 calculated by an unpaired one-way ANOVA, with post-hoc Tukey correction.

The online version of this article includes the following source data for figure 2:

**Source data 1.** Figure 2 - data table.

Hematopoietic lineage cell-specific protein 1 (HS1), the hematopoietic homolog of cortactin (*Huang et al., 2011*), and WASp, the protein defective in Wiskott-Aldrich syndrome (*Bouma et al., 2007*; *Bouma et al., 2011*; *Calle et al., 2004*). To individually assess the role of these two proteins, we used BMDCs cultured from HS1 and WASp knockout mice. As shown in *Figure 2A*, the loss of HS1 had no impact on cortical stiffness of either immature or mature BMDCs. By contrast, mature WASp knockout BMDCs were significantly less stiff than WT controls. This difference mirrors that seen after inhibition of Arp2/3 complex by CK666, suggesting that WASp is the primary activator of Arp2/3 complex-dependent changes in cortical stiffness. The defect in WASp knockout DCs was observed only after maturation; immature WASp knockout DCs did not differ in stiffness from WT controls. This is consistent with our finding that the stiffness of immature DCs is unaffected by actin depolymerizing agents. Taken together, these results show that activation of the WASp-dependent actin polymerization pathway, and to a lesser extent increased myosin contractility and formin-mediated actin polymerization, all contribute to the increased cortical stiffness of mature DCs.

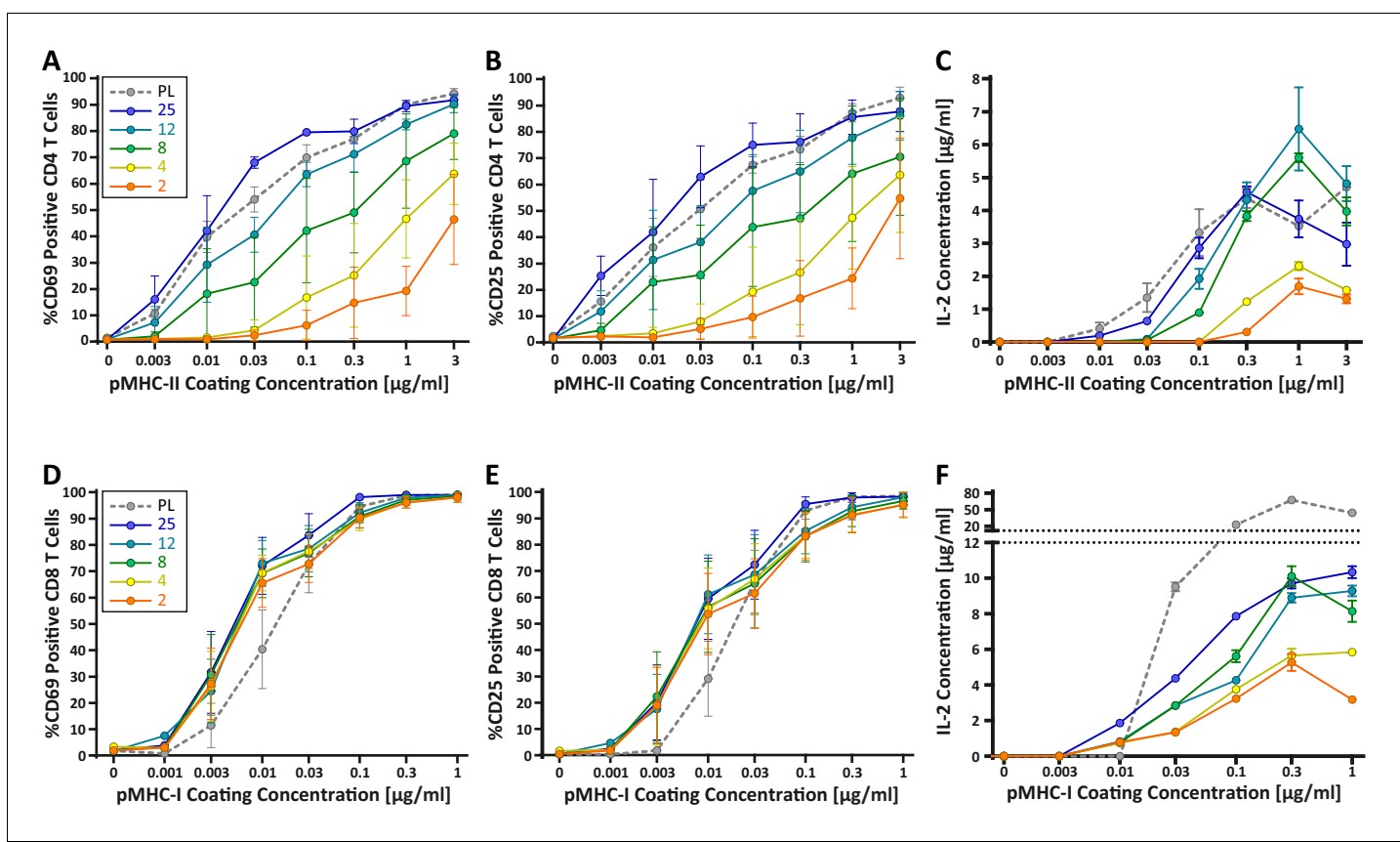

**Figure 3.** CD4[+] and CD8[+] T cells demonstrate vastly different stiffness responses. Murine T cells were purified from lymph nodes and spleen and activated on stimulatory acrylamide hydrogel surfaces with a stiffness range of 2–25 kPa and plastic (PL). Stimulatory surfaces were coated with the indicated concentrations of pMHC-I or pMHC-II together with 2 μg/mL anti-CD28 to stimulate OT-I CD8[+] or OTI-II CD4[+] T cells, respectively. (A,B,D,E) Cells were harvested 24 hr post activation and expression of early activation markers was measured by flow cytometry. Data represent averages +/- SEM of percent positive cells from N = 3 independent experiments. (A,B) CD4[+] T cells show profound stiffness dependent expression of both markers. (C,F) Cell supernatants were collected at 24 hr and IL-2 expression was analyzed by ELISA. Data represent means +/- StDev from three to six replicate samples from one representative experiment, N = 2 experiments.

The online version of this article includes the following source data and figure supplement(s) for figure 3:

**Source data 1.** *Figure 3* - data table.
**Figure supplement 1.** Validation of hydrogel compliances.
**Figure supplement 1—source data 1.** *Figure 3—figure supplement 1* - data table.
**Figure supplement 2.** Binding of pMHC complex to various hydrogel surfaces.
**Figure supplement 2—source data 1.** *Figure 3—figure supplement 2* - data table.

# Stiffness lowers the threshold for activation of CD4$^+$ T cells but not CD8$^+$ T cells

Our findings raise the possibility that changes in DC cortical stiffness, like other maturation-induced changes, enhance the ability of these cells to prime a T cell response. Previous studies have shown that T cells are sensitive to the stiffness of stimulatory surfaces (*Alatoom et al., 2020*; *Judokusumo et al., 2012*; *O'Connor et al., 2012*), and that the TCR serves as the mechanosensor (*Judokusumo et al., 2012*). However, the results are conflicting, and these studies were not performed within the physiological stiffness range that we have defined for DCs. Thus, we tested T cell responses on hydrogels with a stiffness range spanning that of immature and mature DCs (2–25 kPa). Compliance of the hydrogel surfaces was verified by measuring the elastic modulus of the surfaces directly by AFM. Hydrogel stiffnesses were found to be similar to those reported by the manufacturer (*Figure 3—figure supplement 1*). Surfaces were coated with varying doses of peptide-loaded major histocompatibility complex (pMHC) molecules, together with a constant dose of anti-CD28. Surfaces were coated with H-2K$^b$ class I MHC loaded with the N4 (SIINFEKL) peptide (pMHC-I), or I-A$^b$ class II MHC loaded with the OVA$_{329-337}$ (AAHAEINEA) peptide (pMHC-II), to stimulate OT-I CD8$^+$ or OT-II CD4$^+$ T cells, respectively. Plastic surfaces, which are commonly used for stimulation with surface-bound ligands, were included as a familiar reference point. Importantly, the surface chemistry and ligand-binding properties of plastic and hydrogel surfaces are fundamentally different, so direct comparison is not meaningful. To test the effects of substrate stiffness on early T cell activation, we measured surface expression of the activation markers CD25 and CD69, as well as the production of IL-2, all at 24 hr post stimulation. As shown in *Figure 3A–C*, CD4$^+$ T cells showed a profound stiffness-dependent response at 24 hr across all measures. This response was most clearly seen for upregulation of CD25 and CD69, where increasing substrate stiffness enhanced the CD4+ T cell response in a graded manner. As expected, for any given substrate stiffness T cell activation increased with increasing peptide dose. However, comparison among stimulatory surfaces revealed that increasing substrate stiffness lowered the pMHC-II dose required to obtain the same level of activation. Over the stiffness range associated with DC maturation (2–8 kPa), the dose of TCR signal needed to induce surface marker upregulation was shifted by 1–2 logs. Analysis of IL-2 production revealed a similar effect, although the stiffness sensitivity was more bi-modal. IL-2 production increased almost 3-fold when CD4+ T cells were stimulated on surfaces of 8 kPa, as opposed to 2 kPa, for the same antigen dose.

Strikingly, the robust stiffness response we observed in CD4$^+$ T cells was not recapitulated for CD8$^+$ T cells, especially when T cell activation was assessed based on surface marker upregulation (*Figure 3D,E*). Analysis of IL-2 production did reveal stiffness-enhanced activation of naïve CD8$^+$ T cells; as in CD4$^+$ T cells, this was clearest across the stiffness range associated with DC maturation. To determine if the stiffness dependency of CD4$^+$ T cells seen at early times after TCR engagement is maintained at later times, we measured T cell proliferation based on CFSE dilution at 72 hr post stimulation. Similar to what was observed for early activation markers, increasing substrate stiffness produced graded increases in CD4$^+$ T cell proliferation, and the threshold dose required to induce robust proliferation shifted as a function of substrate stiffness (*Figure 4A,B*). This effect was particularly evident at low doses of pMHC-II (0.1–1 ug/mL). Interestingly, although soft hydrogels (2–4 kPa) elicited only very low levels of CD4$^+$ T cell proliferation, these substrates did induce upregulation of CD25 in a high percentage of CD4$^+$ T cells, even in the undivided populations (*Figure 4C,D*). This indicates that an activating signal was received, but was insufficient to drive proliferation.

Since the threshold stimuli (stiffness and dose of pMHC-II) required to induce significant IL-2 production and proliferation were very similar (*Figures 3C* and *4B*), we reasoned that the threshold for proliferation might be driven by IL-2 availability. To test this, OT-II CD4$^+$ T cells were stimulated on hydrogels with or without addition of 25 U/mL exogenous IL-2. Interestingly, addition of IL-2 did not rescue the proliferation of T cells stimulated on soft surfaces, nor did it increase proliferation on stiffer ones (*Figure 4E*). Thus, we conclude that in addition to influencing the signaling threshold for IL-2 production, substrate stiffness also affects other IL-2 independent events needed for efficient T cell proliferation.

Although early activation events in CD8$^+$ T cells showed little to no stiffness sensitivity, CD8$^+$ T cells showed mild stiffness-dependent proliferation (*Figure 5A,B*). The concentration of stimulatory ligand needed to induce at least one round of division was similar across the entire stiffness range

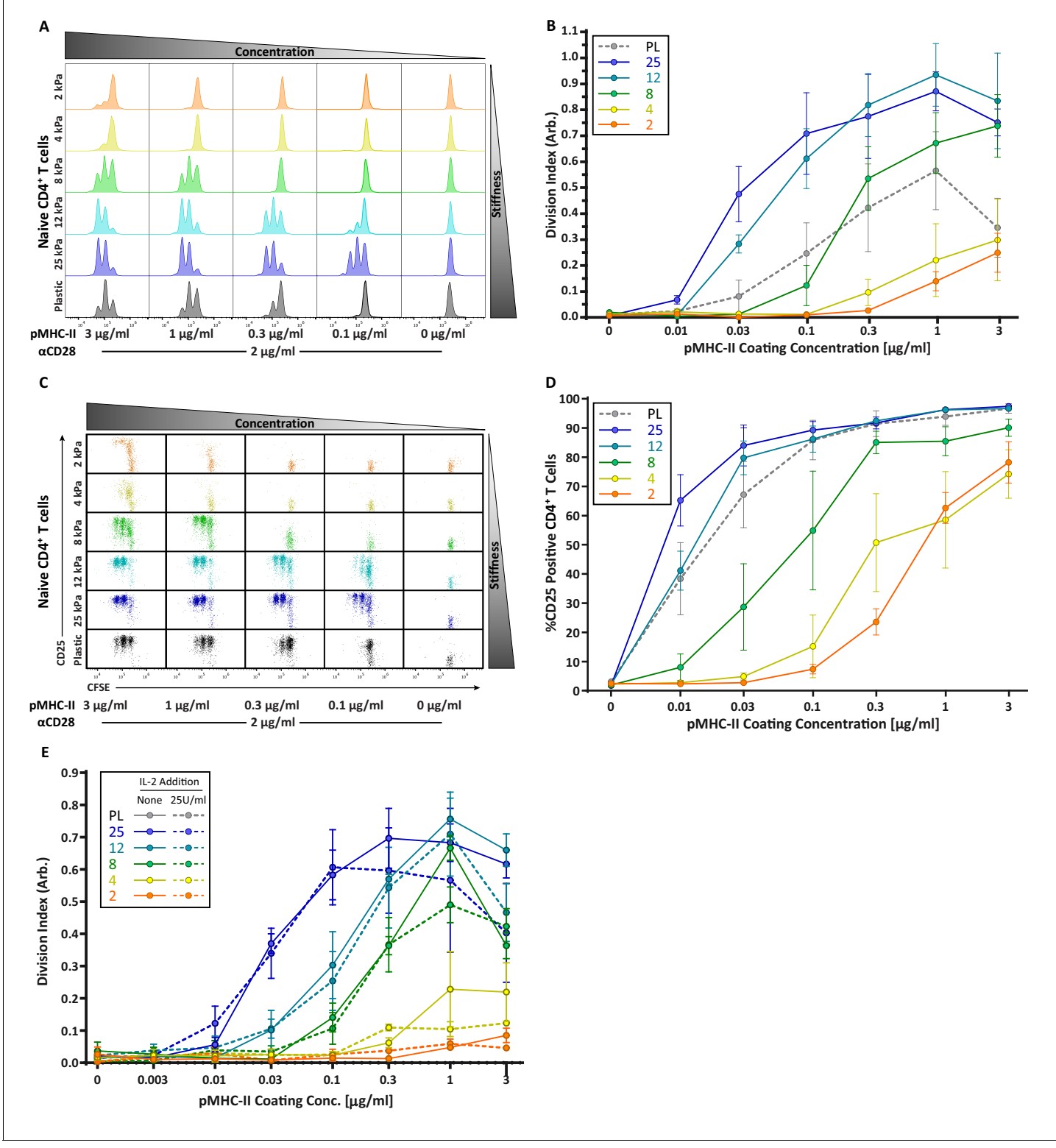

**Figure 4.** CD4 T cell proliferation is stiffness dependent and IL-2 independent. OT-II CD4[+] T cells were purified from lymph nodes and spleen and activated on stimulatory acrylamide hydrogel surfaces with a stiffness range of 2–25 kPa and plastic (PL). Stimulatory surfaces were coated with the indicated concentrations of pMHC-II together with 2 μg/mL anti-CD28. (**A,B**) Proliferation of CD4[+] T cells was measured by CFSE dilution at 72 hr post activation, showing profound stiffness-dependent proliferation. (**A**) Representative CFSE dilution matrix from a single experiment. (**B**) Average division index from three independent experiments. (**C**) Representative plots of CD25 expression as a function of CFSE dilution at 72 hr from a single

*Figure 4 continued on next page*

*Figure 4 continued*

experiment shows that upregulation of CD25 on T cell membrane precedes proliferation. (D) Average percent of T cells expressing CD25 from three independent experiments. (E) Division index of CD4$^+$ T cells activated with or without addition of 25 U/mL of exogenous IL-2. Data in B, D, and E represent averages +/- SEM from at least three independent experiments.

The online version of this article includes the following source data for figure 4:

**Source data 1.** *Figure 4* - data table.

Over successive rounds, increased stiffness did enhance the extent of proliferation, but the differences were relatively small (*Figure 5B*). Interestingly, analysis of CFSE dilution as a function of CD25 expression reveals evidence that CD8$^+$ T cells exhibit a binary stiffness response (*Figure 5C,D*); at low doses of peptide ligand, cells stimulated on very stiff substrates (25 kPa or plastic) survived and proliferated, whereas cells stimulated on softer substrates were mostly lost (*Figure 5C*). From the standpoint of T cell priming, the significance of this observation is unclear as these stiffnesses are well outside the biologically relevant range we measured for DCs.

The observed difference between CD4+ and CD8+ T cell stiffness response could be associated with the difference in antigen strength between the AAHAEINEA (OVA$_{323-339}$) pMHC-II complex used to stimulate CD4$^+$ T cells, and the SIINFEKL (N4) pMHC-I complex used to stimulate CD8$^+$ T cells. To verify that this is not the case, both CD4$^+$ and CD8$^+$ T cells were stimulated in the same way, using surfaces of different stiffness with varying concentrations of αCD3ε together with 2 µg/mL of αCD28. Proliferation was measured based on CFSE dilution at 72 hr post stimulation (*Figure 5—figure supplement 1*). We found that even when both are stimulated through the CD3 complex, CD4$^+$ T cells exhibit profound stiffness sensitivity (especially within the physiological range), while the stiffness responses of CD8$^+$ T cells are more modest. Thus, we conclude that the more robust stiffness response of CD4$^+$ vs CD8$^+$ T cells stems from differences in the cells themselves, rather than the properties of the particular pMHC/TCR pairs used for investigation.

Taken together, our findings point to a mechanism in which stiffer substrates have a sensitizing effect on CD4$^+$ T cells, similar to that of classical costimulatory molecules such as CD28 (*Harding et al., 1992*). When considered in this way, the relative lack of stiffness responses in CD8$^+$ T cells fits with the fact that CD8$^+$ T cells are much less dependent on costimulatory signals (*McAdam et al., 1998*).

## Degranulation of cytotoxic T cells shows mild stiffness sensitivity

Whereas naïve T cells are activated by DCs, effector T cells interact with many cell types. In particular, cytotoxic CD8$^+$ T cells (CTLs) must respond to a variety of possible target cells, which may differ widely with respect to stiffness. We therefore reasoned that CTL effectors might be stiffness independent. To test this, we measured the extent of cytotoxic granule release (degranulation) of effector OTI CTLs stimulated on different stiffness hydrogel surfaces. CTLs were re-stimulated on hydrogels coated with a range of pMHC-I concentrations in the presence of fluorescent anti-CD107a antibody and the amount of CD107a on the cell membrane was analyzed using flow cytometry. Degranulation showed only a mild stiffness dependency (*Figure 5E*), and stiffness tended to affect the magnitude of degranulation rather than whether or not a degranulation response was triggered. Interestingly, changes in substrate stiffness within the range of defined for DCs (2–8 kPa) had little or no impact on degranulation. Increased degranulation was only seen on stiffer surfaces (12 and 25 kPa). This could have functional consequences for effector function in vivo as target cells in inflamed tissues can reach this stiffness range.

## Engagement of the TCR complex by pMHC elicits the most prominent stiffness response

Many current models for TCR mechanosensing are founded on the notion of TCR deformation following the engagement of cognate pMHC (*Ma et al., 2008*). According to this concept, forces applied by the T cell on the TCR-pMHC bond result in conformational changes within TCRαβ (primarily the β chain), which are transmitted to intracellular components of the TCR/CD3 complex, leading to the initiation of downstream signaling (*Das et al., 2015*; *Lee et al., 2015*; *Swamy et al., 2016*). Given this, we wondered whether engaging the TCR complex through the CD3ε chain, as

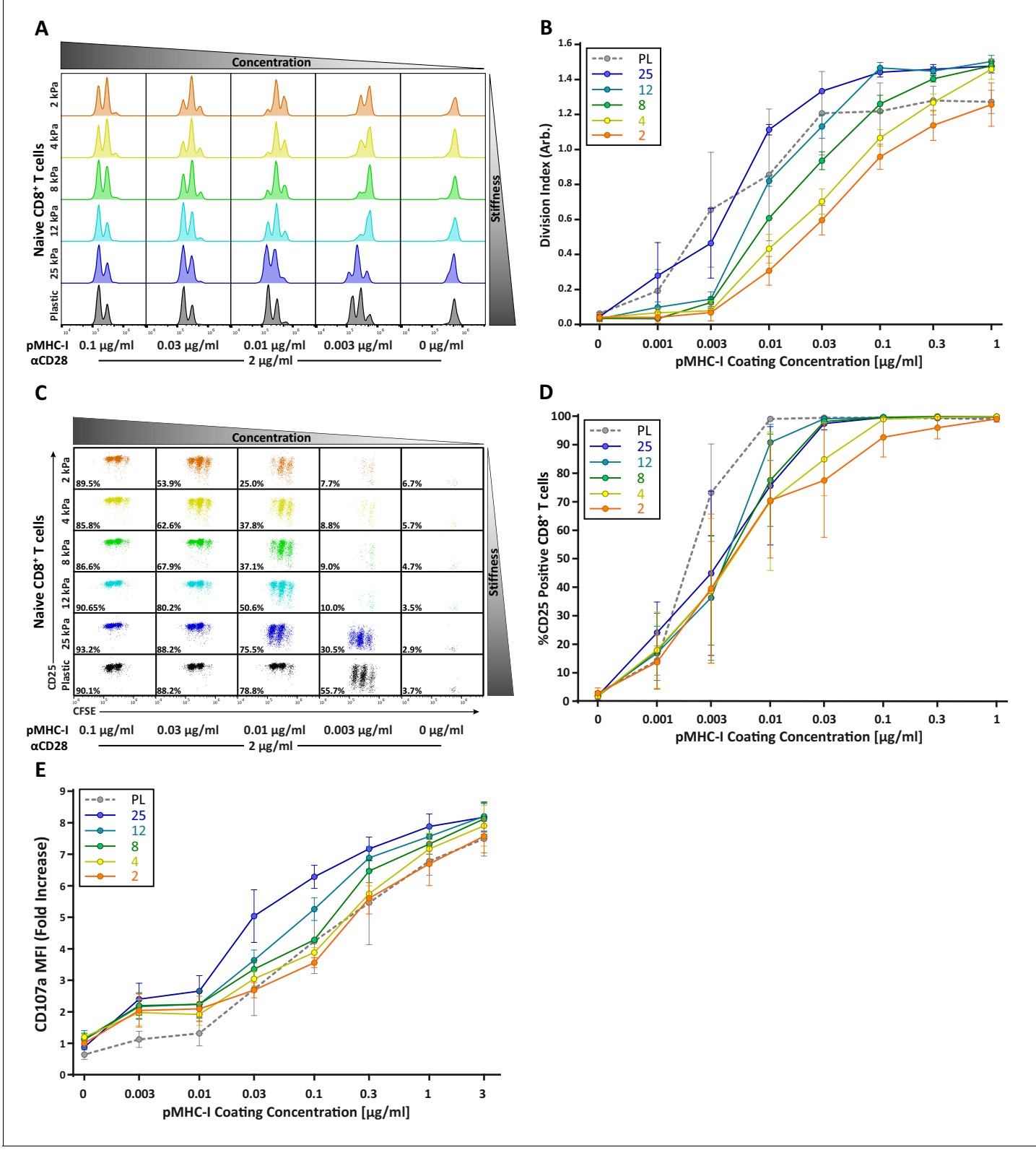

**Figure 5.** CD8[+] T cell proliferation and degranulation show moderate stiffness dependency. Naïve OT-I CD8[+] T cells were purified from lymph nodes and spleen and activated on stimulatory acrylamide hydrogel surfaces with a stiffness range of 2–25 kPa and plastic (PL). Stimulatory surfaces were coated with the indicated concentrations of pMHC-I together with 2 µg/mL anti-CD28. (**A,B**) Proliferation of CD8[+] T cells was measured by CFSE dilution at 72 hr post activation, showing only moderate stiffness-dependent proliferation. (**A**) Representative CFSE dilution matrix from a single

*Figure 5 continued on next page*

*Figure 5 continued*

experiment. Note that the threshold pMHC-I concentration needed to induce proliferation is very similar between the different stiffness surfaces. (B) Average division index from three independent experiments. (C) Representative plot of CD25 expression as a function of CFSE dilution at 72 hr from a single experiment shows a binary stiffness response. Note that with low doses of pMHC-I, only cells stimulated on very stiff substrates (25 kPa or plastic) survived and proliferated. Percent of live cells is given for each condition. (D) Average percent of T cells expressing CD25 from two independent experiments. Levels of CD25 membrane expression are very similar between the different substrates, probably reflecting the fact that only T cells that upregulate CD25 survive. (E) Cytotoxic CD8+ T cells on day 8 or 9 of culture were restimulated on hydrogel surfaces with a range of pMHC-I concentrations, and degranulation was measured based on surface exposure of CD107a (N = 3). Data in B, D, and E represent averages +/- SEM from at least three independent experiments.

The online version of this article includes the following source data and figure supplement(s) for figure 5:

**Source data 1.** *Figure 5* - data table.
**Figure supplement 1.** Comparing T cell activation with 2C11 and pMHC complexes.
**Figure supplement 1—source data 1.** *Figure 5—figure supplement 1* - data table.

compared to direct TCRαβ engagement, would differentially affect T cell mechanosensing. To test this, CD4+ OT-II T cells were stimulated on hydrogels coated with anti-CD3ε antibodies, anti-TCRβ antibodies, or pMHC-II monomers. T cell activation was measured at 72 hr based on proliferation (CFSE dilution) and expression of CD25 (*Figure 6*). All three ligands induced a stiffness-dependent

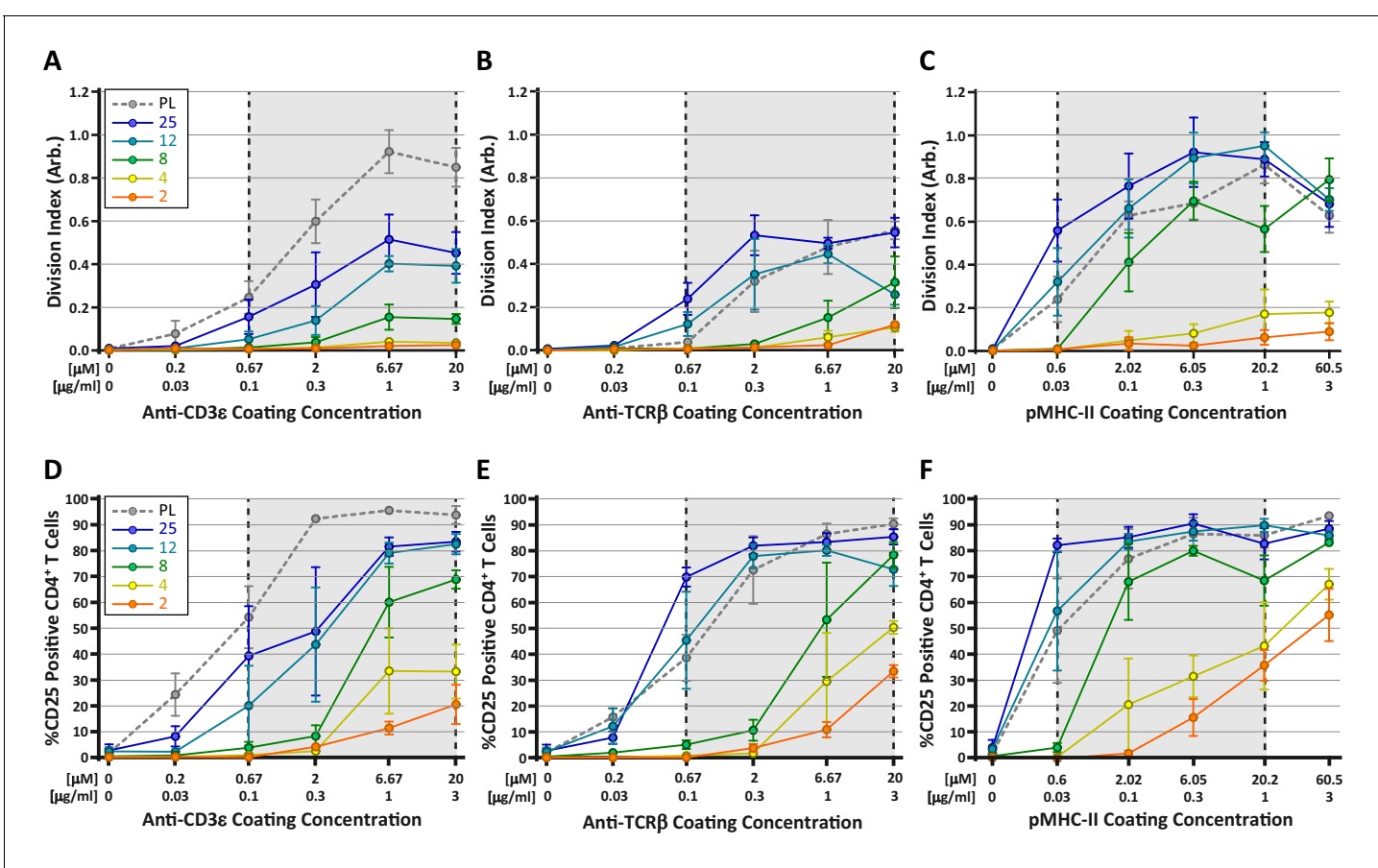

**Figure 6.** Stimulation of TCR with pMHC leads to the strongest stiffness-dependent response. OT-II CD4+ T cells were stimulated on acrylamide hydrogels of different stiffnesses coated with the indicated range of stimulatory ligands. Proliferation and membrane expression of CD25 were measured 72 hr post activation using flow cytometry. (A,D) Stimulation with anti-CD3ε antibody. (B,E) Stimulation with anti-TCRβ antibody. (C,F) Stimulation with pMHC-II. Plots show the average division index or CD25 expression from three independent experiment. Gray areas denote a similar range of stimulatory ligand molar concentrations to aid comparison.

The online version of this article includes the following source data for figure 6:

**Source data 1.** *Figure 6* - data table.

response for both proliferation and CD25 expression, but the responses differed in significant ways. In general, stimulating T cells with pMHC-II resulted in the strongest responses on the hydrogel surfaces; only on plastic surfaces did anti-CD3ε stimulation yield a similarly strong response (*Figure 6A, C*). Importantly, on surfaces with a stiffness similar to that of mature DCs (8 kPa), stimulation with pMHC resulted in robust proliferation, while stimulation with anti-CD3ε yielded a minimal response. On softer substrates (2–4 kPa), pMHC elicited some proliferation, whereas stimulation with anti-CD3 did not. Interestingly, stimulation with anti-TCRβ resulted in a mixed response. At high doses of antigen, stimulation with anti-TCRβ yielded clear proliferative responses on substrates within the biologically-relevant stiffness range. Analysis of CD25 expression patterns revealed a similar trend (*Figure 6D–F*); on soft surfaces, pMHC yielded the strongest response, and anti-TCRβ was more effective than anti-CD3ε. Taken together, these results indicate that T cells sense substrate stiffness best through direct engagement of TCRαβ as predicted by the receptor deformation model.

## Increased stiffness of mature DCs enhances their ability to prime T cells

Our hydrogel assays show that T cell activation is enhanced by changes in stiffness over the range observed for DC maturation, consistent with the idea that modulation of cortical stiffness is a biophysical mechanism by which DCs control T cell activation. To test this directly, we sought conditions under which we could manipulate the stiffness of mature DCs. We took advantage of our finding that mature WASp-KO BMDCs are approximately 20% softer than WT controls (*Figure 2A*; data are presented as absolute values in *Figure 7A*). WT and WASp-KO BMDCs were pulsed with increasing concentrations of OVA$_{323-339}$ peptide and used to prime OT-II CD4$^+$ T cells. As shown in *Figure 7B*, WASp-KO BMDCs did not prime T cells as efficiently as WT BMDCs at low OVA concentrations. Higher concentrations of OVA rescued this defect, showing that loss of WASp shifts the dose of peptide needed rather than affecting T cell priming per se, in keeping with the view that DC stiffness provides a costimulatory signal. Next, we attempted to test T cell priming activity of DCs that are stiffer than WT cells. We tested several genetic manipulations, most of which did not significantly increase the cortical stiffness of mature BMDCs. We did find that overexpression of a constitutively active form of WASp (I294T, CA-WASp [*Beel et al., 2009*]) increased cortical stiffness of mature BMDCs by approximately 20% relative to WT cells (*Figure 7A*), but these BMDCs failed to prime T cells more efficiently (*Figure 7C*). Expression of CA-WASp only enhances BMDC stiffness to approximately 5 kPa, and based on our hydrogel studies, this increase is unlikely to be sufficient to enhance T cell activation. It seems likely that conditions that stiffen DCs to 10 kPa or more would further enhance T cell responses, but we were unable to test this directly, and it is not clear whether this happens in vivo. Nonetheless, the studies using WASp-KO DCs show that changes in DC stiffness impact their ability to efficiently prime a T cell response.

## Discussion

Recent studies have shown that T cell activation involves mechanical cues (reviewed in *Blumenthal and Burkhardt, 2020*). We have previously shown that the DC cytoskeleton constrains the mobility of stimulatory ligands on the DC surface, enhancing T cell activation by opposing the forces exerted by the T cell on the corresponding receptors (*Comrie et al., 2015*). In this study, we elucidate a second mechanism whereby the DC cytoskeleton enhances T cell activation. We show that actin remodeling during DC maturation increases the cortical stiffness of DC by 2–3-fold, and that T cell activation is enhanced by increases in stiffness over the same range. Importantly, increased stiffness lowers the threshold dose of TCR ligand needed for T cell activation, as expected if substrate stiffness serves as a costimulatory signal. In keeping with this concept, CD4$^+$ T cells showed more profound stiffness-sensitivity than CD8$^+$ T cells, especially at early times in the activation process. Together, these results indicate that stiffening of the DC cortex during maturation provides biophysical cues that work together with canonical costimulatory cues to enhance T cell priming.

Modulation of actin architecture has long been appreciated as an essential feature of DC maturation. Changes in the DC actin cytoskeleton facilitate the transition from highly endocytic tissue-resident cells to migratory cells specialized for antigen presentation (*Burns et al., 2004*; *Burns et al., 2001*). Our findings reveal a new facet of this process. We show that immature DCs are very soft, and that upon maturation, their cortical stiffness is increased by 2–3-fold. This is true for cultured

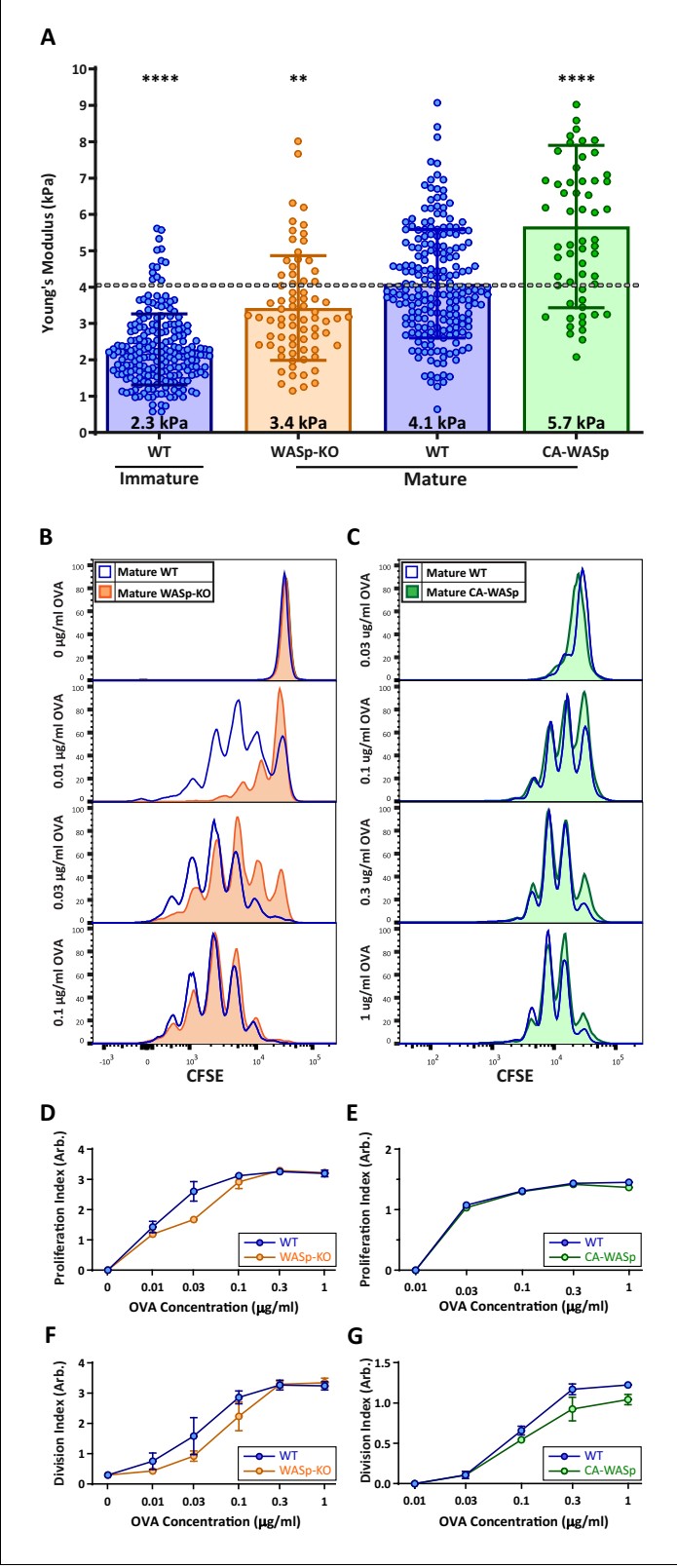

**Figure 7.** DC cortical stiffness acts as a costimulatory signal for T cell activation. WT or WASp[-/-] BMDCs, or WT BMDCs transduced with constituently active form of WASp (CA-WASp) were untreated or matured by treatment with LPS. (**A**) Cortical stiffness was measured by AFM microindentation. Each data point represents an average of two stiffness measurements at different locations around a single cell nucleus. Error bars denote standard

*Figure 7 continued on next page*

*Figure 7 continued*

deviation. \*\*\*p<0.001, \*\*\*\*p<0.0001 calculated by an unpaired one-way ANOVA, comparing mature WT with all other treatments, with post-hoc Tukey correction. (B,C) LPS-matured BMDCs were pulsed with a range of OVA$_{323-339}$ peptide concentrations and co-cultured with ex-vivo OT-II CD4$^+$ T cells for 72 hr. Proliferation was measured by CFSE dilution. (D,E) Proliferation index pooled from two independent experiments. (F–G) Division index values pooled from two independent experiments. Error bars represent StDev.

The online version of this article includes the following source data for figure 7:

**Source data 1.** *Figure 7* - data table.

---

BMDCs treated with lipopolysaccharide in vitro, as well and splenic DCs harvested from LPS-treated mice. A similar trend was reported by *Bufi et al., 2015* for human monocyte-derived DCs, although that study reported lower absolute Young's modulus values. While we used AFM indentation, Bufi et al. used microplate rheology. Since different methods for measuring cell mechanical properties produce absolute Young's modulus values that can vary by as much as 100-fold (*Wu et al., 2018*), it seems likely that the apparent discrepancy in absolute values stems from technical differences between the two studies. Nevertheless, it is clear from both studies that the stiffness of the DC cortex is modulated during maturation.

It is important to note that DCs have complex surface topologies with prominent invaginations and projections that change dramatically during maturation (*Knight et al., 1986*; *Verdijk et al., 2004*). In particular, mature DCs exhibit characteristic membrane veils, as well as microvilli-rich regions that serve as preferred docking sites for T cells (*Fisher et al., 2008*). This complexity makes the interpretation of AFM measurements of cortical stiffness more challenging, as measuring stiffness on a protrusive veil may yield a different result than measuring stiffness directly over the cell body. Because these different structures cannot be resolved by light microscopy we were unable to test for regional stiffness differences (apart from avoiding the nucleus of the cell). Importantly, there were very few instances where measurement of stiffness at two different locations of the same cell resulted in significantly different values (data not shown). Going forward, it will be interesting to determine whether the area of the DC cortex directly underlying an interacting T cell has distinct stiffness properties, and whether this represents a feature of T cell docking sites, or a localized effect of T cell interaction on the DC cytoskeleton.

The observed increase in stiffness depends on changes in actin architecture; whereas depolymerization of actin filaments does not affect on the stiffness of immature DCs, the increase associated with maturation depends on intact filaments, and is sensitive to inhibitors of actin polymerizing molecules. While it remains to be determined exactly which actin regulatory pathways control cortical stiffness in mature DCs, our data show that both Arp2/3 complex and formins are involved. Moreover, we find that DCs lacking the Arp2/3 activator WASp are abnormally soft. In keeping with these findings, DC maturation is known to induce changes in the activation state and localization of Rho family GTPases, especially Cdc42, a molecule that can activate both WASp and formins (*Garrett et al., 2000*; *Vargas et al., 2016*; *West et al., 2000*). Since the overall levels of active Cdc42 are diminished during DC activation, it seems likely that the observed increase in cortical stiffness results from redistribution of the active pool.

We show that DC cortical stiffness is a cell-intrinsic property that is unaffected by substrate stiffness. In this respect, DCs are different from other cell types that adapt their stiffness to differences in substrate compliance (*Byfield et al., 2009*; *Tee et al., 2011*). The ability of DCs to maintain constant stiffness despite changing environmental cues is reminiscent of previous work showing that DCs rapidly change their method of locomotion to maintain consistent migration speed and shape while crossing over different surfaces (*Renkawitz et al., 2009*). This behavior has been proposed to allow DCs to pass through tissues with widely different mechanical properties. In the same way, we propose that the ability of DCs to regulate cortical stiffness as a function of maturation state in spite of environmental cues reflects the importance of this property for priming an appropriate T cell response.

A central finding of this study is that changes in DC stiffness serves as a costimulatory signal for T cell priming. By using a matrix of different hydrogels spanning the biologically relevant range defined for immature and mature DCs (2–8 kPa), coated with increasing pMHC concentrations, we found that stimulatory substrates with higher stiffness required lower concentrations of pMHC to

achieve T cell activation. Similarly, when compared to WT DCs, softer WASp knockout DCs required higher concentrations of OVA peptide to induce the same level of proliferation. Our results indicate that increases in cortical stiffness, together with diminished ligand mobility (*Comrie et al., 2015*), represent biophysical cues that are modulated in parallel with upregulation of costimulatory ligands and cytokines as a fundamental part of DC maturation. When interacting T cells engage pMHC complexes and costimulatory ligands on the DC surface, they integrate this biophysical input along with other canonical costimulatory signals.

In addition to lowering the antigenic threshold for T cell activation, changes in stiffness may present a new signaling mechanism by which DCs control T cell fate and differentiation. *Bufi et al., 2015* showed previously that human monocyte-derived DCs responding to different maturation signals vary in their stiffness. Interestingly, they found that treatment with the tolerizing cytokines TNFα and prostaglandin E2 results in DCs that are even softer than immature cells. Tolerogenic DCs exhibiting partially immature phenotypes have been shown to induce differentiation of regulatory T cells (*Doan et al., 2009*; *Gleisner et al., 2011*; *Gordon et al., 2014*). This effect is usually attributed to low expression of T cell ligands or cytokines, but based on our data, we propose that biophysical properties of the DC cortex also play a role. Going forward, it will be important to ask how DC stiffness is modulated in response to different environmental cues, and whether this further shapes T cell responses.

While we demonstrate T cell stiffness responses on soft surfaces emulating DCs, others have reported T cell stiffness responses on very stiff surfaces (*Judokusumo et al., 2012*; *O'Connor et al., 2012*). We found that very stiff substrates (25 kPa hydrogels and plastic surfaces in the GPa range) elicit strong responses. This was true for proliferation, IL-2 secretion and degranulation. Similarly, recent analysis of human CD4$^+$ effector T cells shows that re-stimulation on soft surfaces induces upregulation of genes related to cytokine signaling and proliferation, while restimulation on very stiff surfaces (100 kPa) triggers expression of an additional genetic program that includes metabolic proteins related to glycolysis and respiratory electron transport (*Saitakis et al., 2017*). The physiological relevance of these augmented responses is unclear as T cells probably never encounter such stiff stimulatory surfaces in vivo. Nonetheless, such findings raise important questions about traditional in vitro assays of T cell function, which often utilize glass or plastic stimulatory surfaces.

The observation that T cells respond to APC stiffness is best understood in the context of evidence that T cells exert force on an interacting APC through the TCR complex (*Bashour et al., 2014*; *Blumenthal and Burkhardt, 2020*; *Hui et al., 2015*; *Husson et al., 2011*; *Li et al., 2010*; *Sawicka et al., 2017*), with the amount of force corresponding to APC stiffness (*Husson et al., 2011*; *Sawicka et al., 2017*). Apart from being a requirement for activation (*Li et al., 2010*; *Pryshchep et al., 2014*), force transduction has been shown to promote peptide discrimination by influencing bond lifetimes (*Das et al., 2015*; *Liu et al., 2014*). Importantly, it appears that the TCR's ability to sense stiffness is closely related to its ability to transduce force-dependent signals during T cell-APC interaction. Indeed, there is evidence that signaling downstream of TCR engagement is increased on stiffer substrates (*Alatoom et al., 2020*; *Judokusumo et al., 2012*) and that the intracellular location of early tyrosine phosphorylation events corresponds to sites of maximum traction force (*Bashour et al., 2014*). We propose that stiffer substrates allow T cells to exert more force through TCR interactions, and consequently induce more effective signaling. This accounts for the costimulatory property of substrate stiffness on T cell activation.

The mechanism by which force application on the TCR is translated into biochemical signals remains controversial. Nevertheless, there is evidence to suggest that force applied on the TCR complex induces conformational changes within TCRαβ that exposes ITAM sites on the CD3 and TCRζ chains for phosphorylation and downstream signaling (*Lee et al., 2015*; *Swamy et al., 2016*) Importantly, conformational changes are mainly attributed to the extension of the CβFG loop region within TCRβ (*Das et al., 2015*), which serves as a lever to push down on the CD3 complex (*Sun et al., 2001*), exposing ITAM sites (*Xu et al., 2008*). In support of this idea, we found that the way in which the TCR is engaged influences T cell stiffness sensing. Within the biologically relevant stiffness range (2–8 kPa), T cells were activated only when TCRαβ was engaged directly; indirect engagement through anti-CD3 resulted in almost no response. We postulate that direct TCRαβ engagement leads to conformational changes that are transmitted appropriately for efficient initiation of downstream signaling, whereas engagement of CD3 induces smaller changes and more limited downstream signaling. This effect may be most evident on soft substrates, because force-dependent

signaling is limiting in this setting. We note that on these soft substrates, pMHC induced stronger T cell activation than anti-TCRβ. This may reflect the involvement of CD4 in the former, which leads to more efficient recruitment of Lck to the TCR complex. One caveat to our work is that pMHC complexes are immobilized on the hydrogel surfaces used here, whereas pMHC complexes show relatively high lateral mobility in the DC membrane, even after maturation (*Comrie et al., 2015*). Future studies addressing how the biophysical properties of the DC surface contribute to tension on the TCR and receptor deformation will need to address the relationship between stiffness and mobility.

Although our focus here is on the role of stiffness sensing in priming of naïve T cells, we find that effector CD8$^+$ T cells also exhibit stiffness-dependent degranulation; stiffness-dependent cytokine production by CTLs has also recently been reported (*Tello-Lafoz et al., 2020*). Similarly, Saitakis et al. have shown that restimulating CD4$^+$ effector T cells on surfaces of different stiffness induces differential gene expression and cytokine production (*Saitakis et al., 2017*). Since DCs increase their cortical stiffness during maturation, a stiffness dependent mechanism for naïve T cell priming makes biological sense. Effector T cells, however, interact with a variety of APCs. In particular, cytotoxic CD8$^+$ T cells are expected to kill any infected cell throughout the body with no stiffness bias. The physiological significance of stiffness sensitivity for effector T cells remains unclear. It is possible that mechanosensing is not needed for effector function per se, but that it is an obligate component of the feedback loop that underlies force-dependent TCR triggering.

The IS is often described as a platform for information exchange between the T cell and APC. Together with our recent work on ligand mobility, the findings presented here indicate that the mechanical properties of the APC side of the IS influence T cell priming, likely because they augment force-dependent conformational changes in TCRs, integrins, and potentially other molecules. Going forward, it will be important to determine how these properties are modulated during DC maturation, and whether there are also local changes induced by signaling events taking place at the IS. In addition, it will be important to tease apart the molecular events through which T cells sense and respond to these mechanical cues.

# Materials and methods

## Key resources table

| Reagent type (species) or resource | Designation | Source or reference | Identifiers | Additional information |
|---|---|---|---|---|
| Antibody | APC/APC-Cy7 anti-CD4 (rat, clone GK1.5) | Biolegend | Biolegend:100411/10041; RRID:100411/AB_312698 | Flow (1:350) |
| Antibody | APC/PE-Cy7 anti-CD8a (rat, clone 53–6.7) | Biolegend | Biolegend: 00711/100721; RRID:AB_312750/AB_312760 | Flow (1:350) |
| Antibody | anti-CD3ε (A. hamster, clone 145–2 C11) | BioXCell | BioXcell: BE0001-1; RRID:AB_1107634 | 0.003 µg/mL - 10 µg/mL |
| Antibody | anti-CD28 (Syrian hamster, clone PV1) | BioXCell | BioXcell:BE0015-5; RRID:AB_1107628 | 1 µg/mL - 2 µg/mL |
| Antibody | Anti-CD86 (rat, clone GL-1) | BioXCell | BioXcell: BE0025; RRID:AB_1107678 | Stain (1:100) |
| Antibody | Alexa647 anti-CD86 (rat, clone GL-1) | BioLegend | BioLegend: 105020; RRID:AB_493464 | Stain (1:100) |
| Antibody | APC-Cy7 anti-CD86 (rat, clone GL-1) | BioLegend | BioLegend:105029; RRID:AB_2074993 | Flow (1:100) |
| Antibody | Anti I-A/I-E (rat, clone M5/114) | BioXCell | BioXcell:BE0108-5; RRID:AB_10949298 | Stain (10 µg/mL) |
| Antibody | PE anti- I-A/I-E (rat, clone M5/114.15.2) | BioLegend | BioLegend:107607; RRID:AB_313322 | Flow (1:100) |
| Antibody | APC anti-CD11c (rat, clone N418) | BioLegend | BioLegend:117309; RRID:AB_313778 | Flow (1:100) |
| Antibody | PE-antiCD107a (LAMP-1) (rat, clone 1D4B) | BioLegend | BioLegend:121611; RRID:AB_1732051 | Assay (2 µg/mL) |

*Continued on next page*

*Continued*

| Reagent type (species) or resource | Designation | Source or reference | Identifiers | Additional information |
|---|---|---|---|---|
| Antibody | Alexa680 goat anti-Rat IgG (H+L) | ThermoFisher | ThermoFisher: A-21096 RRID:AB_2535750 | Stain (1:500) |
| Chemical compound, drug | Cytochalasin-D | EMD Millipore | EMD Millipore: 250255; CAS:22144-77-0 | 10 µM |
| Chemical compound, drug | Latrunculin-B | EMD Millipore | EMD Millipore: 428020; CAS:76343-94-7 | 10 µM |
| Chemical compound, drug | (S)-nitro-Blebbistatin | Cayman Chemical | Cayman Chemical:13891; CAS:856925-75-2 | 50 µM |
| Chemical compound, drug | CK666 | EMD Millipore | Millipore: 182515; CAS:442633-00-3 | 100 µM |
| Chemical compound, drug | Y27632 | SIGMA | SIGMA:688000; CAS:146986-50-7 | 25 µM |
| Chemical compound, drug | SMIFH2 | SIGMA | SIGMA:344092; CAS:340316-62-3 | 10 µM |
| Chemical compound, drug | *Escherichia coli* 026:B6; LPS | SIGMA | SIGMA:L2762; ECN:297-473-0 | 200 ng/mL |
| Chemical compound, drug | Collagenase D | SIGMA | SIGMA: COLLD-RO; EC#:3.4.25.3 | 2 mg/mL |
| Chemical compound, drug | IL-2 | NIAID, NIH | N/A | 25U;100U |
| Chemical compound, drug | streptavidin-coated polystyrene beads | Spherotech | Spherotech: SVP-60–5 | |
| Chemical compound, drug | EZ-link NHS biotin kit | Thermo Fisher Scientific | Thermo Fisher Scientific: 21217 | |
| Chemical compound, drug | CFSE | Invitrogen | Invitrogen: C34570 | |
| Commercial assay or kit | CF555 Mix-n-Stain | Biotium | Biotium:92234 | |
| Commercial assay or kit | MACS pan-dendritic cell isolation kit | Miltenyi Biotec | Miltenyi: 130-100-875 | |
| Commercial assay or kit | mouse IL-2 ELISA kit | Invitrogen | Invitrogen: 88-7024-88 | |
| Gene (mouse) | *Was*; Wiskott–Aldrich syndrome gene | DOI:10.1084/jem.20091245 | MGI:105059; NCBI Gene: 22376 | |
| Other | Hydrogel surfaces (96-well plates) | Matrigen | EasyCoat Softwell 96G | Customized plates |
| Other | AFM 1 µM spherical polystyrene probe | Novascan | Novascan: PT.PS | $Si_3N_4$ cantilever k = 0.06 N/m |
| Peptide, recombinant protein | Peptide MHC-II Complex | NIH Tetramer Core | I-A$^b$ | Sequence: HAAHAEINEA |
| Peptide, recombinant protein | Ovalbumin 323–339; OVA$_{323-339}$ | Anaspec | Anaspec: AS-27025; LOT:1755317 | Sequence: ISQAVHAAHAEINEAGR |
| Recombinant DNA reagent | pLX301 (plasmid) | Addgene | Addgene: 25895; RRID:Addgene_25895 | DOI:10.1038/nmeth.1638 |
| Software, algorithm | FlowJo | FlowJo LLC | RRID:SCR_008520 | |
| Software, algorithm | NanoScope Analysis | Broker | N/A | |
| Strain, strain background (Mice) | OT-II Transgenic mice/OT-II | Jackson Laboratories | Stock: 004194; RRID:IMSR_JAX:004194 | B6.Cg-Tg(TcraTcrb)425Cbn/J |
| Strain, strain background (Mice) | OT-I Transgenic mice/OT-I | Jackson Laboratories | Stock: 003831; RRID:IMSR_JAX:003831 | C57BL/6-Tg(TcraTcrb)1100Mjb/J |

*Continued on next page*

*Continued*

| Reagent type (species) or resource | Designation | Source or reference | Identifiers | Additional information |
|---|---|---|---|---|
| Strain, strain background (Mice) | *Was*<sup></sup>; WASP-,WASp-KO | Jackson Laboratories | Stock: 003292; RRID:IMSR_JAX:003292 | 129S6/SvEvTac-Was<sup>tm1Sbs</sup>/J |
| Strain, strain background (Mice) | *Hcls1*<sup>-/-</sup>; HS1-KO | David Rawlings, MD. | PMCID:PMC394441 | |
| Strain, strain background (Mice) | *Fscn1-/-* | KOMP Repository, UC Davis | MGI:5605764 | Fscn1<sup>tm1.1(KOMP) Vlcg</sup> |

## Inhibitors and antibodies

Cytochalasin-D and latrunculin-B were obtained from EMD Millipore, (S)-nitro-Blebbistatin was purchased from Cayman Chemical, CK666 was from Calbiochem, and Y27632 and SMIFH2 were from Sigma-Aldrich. Flow cytometry antibodies: rat anti-CD4 APC/APC-Cy7 (clone RM4-5), rat anti-CD8a APC/PE-Cy7 (clone 53–6.7), Armenian hamster anti-CD69 APC (clone H1.2F3) and rat anti-CD25 PE (clone PC61) were all purchased from BioLegend. Surface coating antibodies: Armenian hamster anti-CD3ε (clone 2C11) and Armenian hamster anti-CD28 (Clone PV1) were from BioXCell. Biotinylated Armenian hamster anti-CD3ε (clone 2C11) was from Invitrogen, and biotinylated mouse anti-TCRVβ5.1/5.2 (clone MR9-4) was from BD Bioscience. Dendritic cell staining: anti-CD86 CF555 was made by conjugating purified rat anti-CD86 (BioXCell) with CF555 conjugated dye from Biotium as per the manufacturer's protocol.

## Mice

All mice were originally obtained from The Jackson Laboratory and housed in the Children's Hospital of Philadelphia animal facility, according to the guidelines put forth by the Institutional Animal Care and Use Committee. C57BL/6 mice (WT) were purchased from Jackson Labs. HS1-KO mice on the C57BL/6 background have been previously described (*Taniuchi et al., 1995*) and were a kind gift from Dr. David Rawlings at the University of Washington. *Was*<sup>-/-</sup> mice were purchased from Jackson Labs (*Snapper et al., 1998*) and fully backcrossed to a C57BL/6 background. All mouse strains were used as a source of bone marrow from which to generate BMDCs. Mice bearing a gene trap mutation in the *Fscn1* gene (Fscn1<sup>tm1(KOMP)Vlcg</sup>), which abrogates the expression of the protein Fascin 1, were generated by the KOMP Repository at UC Davis, using C57BL/6 embryonic stem cells generated by the Texas A & M Institute for Genomic Medicine. Because these mice proved to have an embryonic lethal phenotype, fetal liver chimeras were used as a source of bone marrow precursors. Heterozygous mating was performed, and fetal livers were collected after 15 days of gestation and processed into a single-cell suspension by mashing through a 35 μm filter. Embryos were genotyped at the time of harvest. Cells were resuspended in freezing media (90% FCS, 10% DMSO) and kept at −80°C until used. Thawed cells were washed, counted, resuspended in sterile PBS and injected intravenous into sub-lethally irradiated 6-week-old C57BL/6 recipients, $1 \times 10^6$ cells per mouse. Chimeras were used as a source for fascin KO bone marrow ~6 weeks after transfer. OT-I T cells were prepared from heterozygous OT-I TCR Tg mice, which express a TCR specific for ovalbumin 257–264 (amino acid sequence SIINFEKL) presented on H-2K<sup>b</sup> (*Hogquist et al., 1994*). OT-II T cells were prepared from heterozygous OT-II TCR Tg mice, which express a TCR specific for ovalbumin 323–339 (amino acid sequence ISQAVHAAHAEINEAGR) presented on I-A<sup>b</sup> (*Barnden et al., 1998*).

## Cell culture

Unless otherwise specified, all tissue culture reagents were from Invitrogen/Life Technologies. GM-CSF was produced from the B78H1/GMCSF.1 cell line (*Levitsky et al., 1996*). HEK-293T cells (ATCC) were cultured in DMEM supplemented with 10% FBS, 25 mM Hepes, penicillin/streptomycin, GlutaMAX, and non-essential amino acids.

Generation of bone marrow-derived dendritic cells (BMDCs) was similar to *Inaba et al., 1992*. Briefly, mouse long bones were flushed with cold PBS, the resulting cell solution was passed through a 40 μm strainer, and red blood cells were lysed by ACK lysis. Cells were washed once with RPMI-1640 and then either frozen for later use in RPMI-1640 containing 20% FBS, 10% DMSO, or plated in 10 cm bacterial plates in BMDC culture media (RPMI-1640, 10% FBS, penicillin/streptomycin,

GlutaMax and 1% GM-CSF supernatant). On day 3 of culture, dishes were supplemented with 10 mL of BMDC culture media. On day 6, 10 mL of media were replaced, by carefully collecting media from the top of the dish and slowly adding fresh media. BMDC differentiation was verified using flow cytometry, showing 80–90% CD11c positive cells. BMDC maturation was induced on day 7 or 8; immature BMDCs were harvested and replated on a 6 cm tissue culture dish in 5 mL of BMDC media supplemented with 200 ng/mL lipopolysaccharide (*Escherichia coli* 026:B6; Sigma-Aldrich) for at least 24 hr. Maturation was verified using flow cytometry, with mature BMDCs defined as Live/CD11c+/CD86high/MHC-IIHigh cells. To generate splenic DCs, spleens from C57BL/6 mice were cut into smaller pieces and digested with collagenase D (2 mg/mL, Sigma) for 30 min at 37°C, 5% $CO_2$. Cells were washed and labeled for separation by negative selection using a MACS pan-dendritic cell isolation kit (Miltenyi Biotec).

Primary mouse T cells were purified from lymph nodes and spleens using MACS negative selection T cell isolation kits (Miltenyi Biotec). In the case of CD4+ T cells, ex vivo cells were used. Since isolation yielded mostly naïve cells (>90%, data not shown), we refer to them as naïve CD4+ T cells. In the case of CD8+ T cells, approx. 45% of T cells isolated from OT-I mice showed some level of activation. Thus, we specifically isolated naïve T cells by MACS purification. To generate cytotoxic CD8+ T cells (CTLs), purified murine CD8+ cells were activated on 24-well plates coated with anti-CD3ε and anti-CD28 (2C11 and PV1, 10 µg/mL and 2 µg/mL, respectively) at $1 \times 10^6$ cells per well. After 24 hr, cells were removed from activation and mixed at a 1:1 vol ratio with complete T cell media (DMEM supplemented with penicillin/streptomycin, 10% FBS, 55 µM β-mercaptoethanol GlutaMAX, and non-essential amino acids), containing recombinant IL-2 (obtained through the NIH AIDS Reagent Program, Division of AIDS, NIAID, NIH from Dr. Maurice Gately, Hoffmann - La Roche Inc [*Lahm and Stein, 1985*]), to give a final IL-2 concentration of 100 units/mL. Cells were cultured at 37°C and 10% $CO_2$, and passaged as needed to be kept at $0.8 \times 10^6$ cells/mL for 7 more days. CTLs were used at day 8 or 9 after activation.

## Plasmid construction, viral production, and transduction of DCs

A constitutively active form of WASp (CA-WASp) was engineered by subcloning WASp cDNA into a pLX301 vector (Addgene), introducing an I294T point mutation (*Westerberg et al., 2010*) by site-directed mutagenesis, and confirming by sequencing. To generate recombinant lentivirus, HEK-293T cells were co-transfected using the calcium phosphate method with psPAX2 and pMD2.G, together with the DNA of interest in pLX301. For transduction, BMDCs were plated in untreated six well plates at $2 \times 10^6$ cells/well in 3 mL of BMDC media. BMDC transduction was carried out on day 2 of culture; lentiviral supernatants were harvested from HEK-293T cells 40 hr post transfection, supplemented with 8 µg/mL polybrene (Sigma-Aldrich), and used immediately to transduce BMDCs by spin-infection at 1000×g, 37°C for 2 hr. After resting the cells for 30 min at 37°C, 5% $CO_2$, lentivirus-containing media was replaced with normal BMDC culture media. On day 5 of culture, puromycin (Sigma-Aldrich) was added to a final concentration of 2 µg/mL to allow selection of transduced BMDCs. Maturation of transduced cells was induced on day 8 by adding 200 ng/mL of lipopolysaccharide in puromycin-free media.

## Flow cytometry

All cells were stained with Live/Dead aqua (ThermoFisher) following labeling with appropriate antibodies in FACS buffer (PBS, 5% FBS, 0.02% $NaN_3$, and 1 mM EDTA). Flow cytometry was performed using either the Cytoflex LX or CytoFlex S cytometer (Beckman Coulter) and analyzed using FlowJo software (FlowJo LLC). T cells were gated based on size, live cells, and expression of CD4 or CD8 (depending on the experiment). DCs were incubated for 10 min on ice with the Fc blocking antibody 2.4G2 before staining. DCs were gated based on size, live cells, and CD11c expression. Mature DCs were further gated based on high expression of MHC-II, CD86, or CD80.

## T cell activation on stimulatory gel surfaces

96-well plates coated with polyacrylamide hydrogels spanning a stiffness range of 2–25 kPa were obtained from Matrigen. Hydrogel stiffness was verified by AFM at different locations around the hydrogel surface (*Figure 3—figure supplement 1*). Surfaces were first coated with 10 µg/mL of NeutrAvidin (ThermoFisher) and 2 µg/mL anti-CD28 (clone PV1) overnight at 4°C. Primary amines in the

NeutrAvidin form covalent bonds with quinone functional groups within the hydrogels. The gel pore size is on the order of tens of nanometers, such that cells can only interact with ligands bound on the gel surface. Surfaces were then washed twice with 200 µL of PBS and coated with varying concentrations of biotinylated pMHC monomers (NIH tetramer core facility) for 2 hr at 37°C. In cases where antibody stimulation was compared to pMHC stimulation, surfaces were coated with varying concentrations of either biotinylated anti-CD3ε (clone 2C11), or biotinylated anti-TCRvβ5.1/5.2 (clone MR9-4). In experiments where only antibody stimulation was used, surfaces were coated directly with varying concentrations of anti-CD3ε (clone 2C11) together with 2 µg/mL anti-CD28 (clone PV1) for 2 hr at 37°C. Following coating, surfaces were washed two times with 200 µL of PBS, and blocked for 10 min with T cell media containing 10% FBS before addition of $2.0 \times 10^5$ cells/well. Control studies showed that stimulatory ligands bound slightly less well to stiffer hydrogel surfaces (*Figure 3—figure supplement 2*). Stiffer surfaces show the same or higher activating properties across all assays, ruling out the possibility that differences in T cell activation are due to differential ligand binding. Importantly, initial experiments included a 1 kPa hydrogel surface that yielded no response across all assays. Therefore, data from this condition is not shown and was not included in repeated experiments. For experiments where exogenous IL-2 was added, media was supplemented with IL-2 to a final concentration of 25 U/mL. For measurements of CD69/CD25 expression and IL-2 production, cells were harvested 22–24 hr post stimulation for flow cytometry, and supernatants were used to measure IL-2 concentration using a mouse IL-2 ELISA kit (Invitrogen). For early activation marker expression assays, cells were plated immediately after isolation, and harvested 22–24 hr post stimulation for flow cytometry analysis. For CFSE dilution assays, purified cells were washed once with PBS and stained for 3 min with 2.5 µM CFSE (ThermoFisher). After quenching the excess CFSE by addition of 1 mL FBS for 30 s, cells were washed and plated. Cells were harvested 44–48 hr ($CD8^+$ T cells) or 68–72 hr ($CD4^+$ T cells) post stimulation for flow cytometry analysis. For ligand comparison assays, surfaces were first coated with 10 µg/mL of NeutrAvidin (Thermo Fisher) and 2 µg/mL anti-CD28 (PV1) overnight at 4°C. Surfaces were then washed twice with 200 µL of PBS and coated with varying concentrations of biotinylated ligands (anti-TCRVβ5.1/5.2, anti-CD3ε, or pMHC-II monomers) for 2 hr at 37°C.

## Cytotoxic T cell degranulation assays

Assays were conducted on day 8 or 9 of culture. About $2 \times 10^5$ CTLs were plated onto surfaces coated with various concentrations of pMHC-I in the presence of 2 µg/mL PE-conjugated anti-CD107a. After 3 hr of re-stimulation, CD107a labeling was quantified using flow cytometry . Cells were gated based on size, live cells, and expression of $CD8^+$. CD107a mean fluorescence intensity (MFI) was extracted using FlowJo.

## Hydrogel ligand-binding assays

Hydrogel surfaces were coated with 1 µg/mL pMHC-II monomers as described earlier. Hydrogel wells were washed thrice with 200 µL of PBS and blocked for 1 hr with 0.25% bovine gelatin in PBS solution. After blocking, pMHC-II molecules were stained with 10 µg/mL rat anti-mouse I-A/I-E antibody (Clone M5/114) in 0.25% bovine gelatin in PBS solution for 1 hr at room temperature. Wells were than washed thrice and stained with Alexa 680 conjugated goat anti-rat secondary antibody diluted 1:500 in 0.25% bovine gelatin in PBS solution for 1 hr at room temperature. Finally, wells were washed and imaged using a Licor Odyssey CLx reader.

## T cell priming assays

Priming assays were carried out in round bottom 96-well plates. About $5 \times 10^4$ LPS-matured BMDCs were plated in each well and pulsed with OVA$_{323-339}$ peptide at various concentrations (0.1–1 µg/mL). $1.5 \times 10^5$ CFSE stained, OT-II $CD4^+$ T cells were added to each well and incubated for 68–72 hr. Cells were then harvested and analyzed using flow cytometry.

## Atomic force microscopy

All experiments were carried out at room temperature using a Bruker Bioscope Catalyst AFM mounted on a Nikon TE200 inverted microscope. Microindentation measurements were made with a spherical tip from Novascan. The tip was comprised of a 1 µm silicon dioxide particle mounted on a

silicon nitride cantilever with a nominal spring constant of 0.06 N/m; each cantilever was calibrated using the thermal fluctuation method. The AFM was operated in fluid contact mode, with 2 Hz acquisition. Total vertical cantilever displacement was set to 5 μm, producing a maximal approach/retraction speed of ~20 μm/s. Maximal deflection (Trigger threshold) was adjusted for each cantilever to apply a maximal force of 6 nN on the measured cell (e.g. for a 0.06 N/m cantilever, the trigger threshold was set to 100 nm). The actual indentation depth was ~1.5 μm depending on the measured cell stiffness (*Figure 1—figure supplement 4A, B*). Analysis of force-distance curves was carried out using the Nanoscope Analysis software (Bruker). The Young's modulus was extracted using the Hertzian model for spherical tips with a contact point-based fitting on the extend curve data. Importantly, one of the Hertzian model requirements is that indentation depth does not exceed the radius of the spherical tip. Since the Bruker software we use does not allow one to restrict the fitting algorithm based on Z axis displacement, we were unable to restrict fitting to exactly 0.5 μm. Instead, we restricted the Hertzian model fitting to 30% of total force applied, which we found corresponds to ~0.5 μm indentation depth (*Figure 1—figure supplement 4C*). For each individual cell, two separate measurements were conducted at different locations near but not directly over the nucleus. The reported cell stiffness value represents the average between these independent measurements. Note that when measurements of cortical stiffness were made over the nucleus, no significant differences in Young's modulus values were found (not shown). To measure BMDC stiffness, $1 \times 10^5$ cells (untreated or lipopolysaccharide matured) were seeded onto poly L-lysine coated coverslips and allowed to spread for 4 hr at 37°C, 5% $CO_2$. Before data acquisition, cells were incubated for 10 min with the Fc blocking antibody 2.4G2, washed and stained for CD86 for 20 min, then washed and mounted on the AFM. All antibody incubations and data acquisition steps were performed in L-15 media (Gibco) supplemented with 2 mg/mL glucose. For treated cell measurements, drugs [latrunculin-B (10 μM), cytochalasin-D (10 μM), s-nitro-blebbistatin (50 μM), Y27632 (25 μM), CK666 (100 μM), or SMIFH2 (10 μM)] were pre-incubated with the cells at 37°C, 5% $CO_2$ for 30 min before Fc blocking and maintained in the cultures throughout staining and data acquisition. Measured cells were visually selected based on fluorescence; immature/mature cells were distinguished based on CD86 staining. In experiments where the GFP-CA-WASp construct was expressed, GFP positive cells were selected in conjunction with CD86 staining using a dual-band fluorescence filter set.

## Dendritic cell imaging

About $1 \times 10^5$ DCs (untreated or lipopolysaccharide matured) either from WT or GFP-Lifeact Tg mice were seeded onto poly L-lysine coated hydrogels or glass coverslips and allowed to spread for 4 hr at 37°C, 5% $CO_2$. Before imaging, cells were incubated for 10 min with the Fc blocking antibody 2.4G2, washed and stained with Alexa647 conjugated anti-CD86 (Clone GL-1) for 20 min, then washed and mounted on the microscope. Antibody incubations and data acquisition steps were performed in L-15 media (Gibco) supplemented with 2 mg/mL glucose. Imaging DCs on hydrogels was done using a 40X, long working distance objective through the hydrogels.

## Statistical methods

All datasets were subjected to outlier analysis before execution of statistical testing. Outliers were defined as data points with values outside the range of mean +/- 2.5 xStDev and were deleted from the dataset. Testing for a statistically significant difference between experimental groups was done using an unpaired one-way ANOVA test with a post-hoc Tukey correction for multiple comparisons.

Throughout the paper, data shown represents biological, and not technical, replicates. For BMDC assays, a single experiment constitutes measurement of multiple cells from a fresh DC culture, starting from frozen or freshly harvested bone marrow. For splenic DCs, a single experiment constitutes measurement of multiple cells freshly purified from the spleen of a single mouse. In each experiment, WT or untreated cells were measured side by side with treated cells as a standard control. For T cell assays, a single experiment constitutes cells freshly purified from spleen and lymph nodes of a single animal. All CFSE dilution assays were executed in technical duplicates, although a single data set is presented. When needed, figure legends describe the quantity of technical repeats used in an experiment.

## Acknowledgements

The authors thank Florin Tuluc and Jennifer Murray from the Children's Hospital of Philadelphia Flow Cytometry core. We thank the biomechanics core of the Institute of Translational Medicine and Therapeutics (ITMAT) at the University of Pennsylvania for use of the Atomic Force Microscope and the NIH tetramer core facility for provision of MHC-peptide complexes. We thank Dr. Shuixing Li and Dr. Nathan Roy for expert technical assistance, Dr. Nathan Roy and Mr. Tanner Robertson for critical reading of the manuscript, and members of the Burkhardt laboratory for many helpful discussions. This work was supported by NIH grants R01 GM104867 and R21 AI32828 to JKB.

## Additional information

### Funding

| Funder | Grant reference number | Author |
| --- | --- | --- |
| National Institute of General Medical Sciences | GM104867 | Janis K Burkhardt |
| National Institute of Allergy and Infectious Diseases | AI32828 | Janis K Burkhardt |

The funders had no role in study design, data collection and interpretation, or the decision to submit the work for publication.

### Author contributions

Daniel Blumenthal, Conceptualization, Data curation, Formal analysis, Supervision, Validation, Investigation, Methodology, Writing - original draft, Writing - review and editing; Vidhi Chandra, Lyndsay Avery, Formal analysis, Investigation; Janis K Burkhardt, Conceptualization, Resources, Supervision, Funding acquisition, Methodology, Writing - review and editing

### Author ORCIDs

Daniel Blumenthal (iD) https://orcid.org/0000-0002-9207-5848
Vidhi Chandra (iD) http://orcid.org/0000-0003-3642-2144
Lyndsay Avery (iD) https://orcid.org/0000-0002-8148-1648
Janis K Burkhardt (iD) https://orcid.org/0000-0002-8176-1375

### Ethics

Animal experimentation: All studies, breeding and maintenance of animals was performed under Animal Care and Use Protocol #667, as approved by The Children's Hospital of Philadelphia Institutional Animal Care and Use Committee.

### Decision letter and Author response

Decision letter https://doi.org/10.7554/eLife.55995.sa1
Author response https://doi.org/10.7554/eLife.55995.sa2

## Additional files

### Supplementary files

• Transparent reporting form

### Data availability

All data generated or analysed during this study are included in the manuscript and supporting files.

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
