## [Decision Letter]

**Acceptance summary:**

Your carefully conducted study is particularly interesting since it represents a coherent and complete analysis of the response (expression of activation markers, IL-2 production and proliferation) of mouse T cells to stiffness. It also demonstrates that the program of maturation of DC includes changing in their mechanical properties that is most probably involved in their ability to activate T lymphocytes. Your work also open new fields of investigation for example: what are the mechanisms involved in the modification of the DC cytoskeleton leading to changes in their stiffness? What are the molecular mechanisms, which explain the difference of sensitivity to stiffness between CD4 and CD8 T cells?

**Decision letter after peer review:**

Thank you for submitting your article "T-cell priming is enhanced by maturation-dependent stiffening of the dendritic cell cortex" for consideration by *eLife*. Your article has been reviewed by three peer reviewers, including Michael L Dustin as the Reviewing Editor and Reviewer #1, and the evaluation has been overseen by a Reviewing Editor and Michel Nussenzweig as the Senior Editor. The following individuals involved in review of your submission have agreed to reveal their identity: Claire Hivroz (Reviewer #3).

The reviewers have discussed the reviews with one another and the Reviewing Editor has drafted this decision to help you prepare a revised submission.

Summary:

This is an informative study that addresses an important topic related to the stiffness of dendritic cells and response to T cells to stimuli provided on artificial surfaces with bulk stiffness in this range. This is an important area in immunology and biophysics. Key findings include that DC increase in stiffness during maturation and the stiffness is nearly 5-fold greater than the previous best measurements. T cells respond better to anti-CD3 antibodies presented on gels of similar stillness to mature DC than to stiffness of immature DC. Response to anti-CD28 is not sensitive to stiffness. Peripheral blood T cells are not sensitive in this range of stiffness. DC don't become stiffer on a stiffer substrate like other cells, which makes it hard to experimentally manipulate their stiffness. WASp contributes to DC stiffness and WASp deficient DC are less stimulatory. However, stiffer active WASp expressing DC do not stimulate T cells better. There are strengths and weaknesses to this study. The most important major issues are addressed below.

Essential revisions:

1) AFM measurements – Earlier measurements from Bufi et al., 2015 showed the same trend as here, but lower values. P.-H. Wu et al., 2018 shows that AFM probes of different dimensions generate different stiffness values with smaller probes giving higher values of Young's modulus. Since Bufi et al., used an infinitely large probe (a flat paddle) and this study used a 1 µm bead as the probe that larger value measured here may be a consequence of this and this should be discussed. The gels that are used for the T cell activation experiments are commercial and their stiffness is presumably specified by the supplier. The authors should measure the stiffness of these gels under the same conditions as the cells and determine the measured stiffness, which will ensure that the gels truly replicate the cellular stiffness. There are also some details of the measurements that should be better described. The indentation length cannot be of 5 μm since this is more the tip size (1 μm bead)…I think this is probably the range of vertical displacement of the cantilever, the motion of the cantilever being stopped when the force it is facing reaches 6 nN. As far as I understand, this indicates a cantilever deflection of 0.1 μm (100 nm), but does not say anything about the indentation depth in the cell, and this is important information to evaluate to what extent it is possible to claim that the authors have only probed the cortex of the cells. Finally, the authors should also explain the meaning of the 2 Hz acquistion frequency. Contact mode measurements are not supposed to be sine waves. So, I think this is the frequency at which measurements were repeated. It would be better to indicate the approach/retraction rate, i.e. a speed in μm/s. This parameter has indeed an important impact on the measured stiffness value. The authors should also provide information about where on the cell they were indenting and if the location mattered?

2) The experiments to manipulate DC stiffness are important to the impact of the paper beyond earlier work. It’s interesting that mature DC lacking WASp are both softer and less potent in T cell stimulation, but there may be many other explanations for this. The disappointing result that CA-WASp didn't further increase T cell responses might reflect saturation of the response to stiffness. Can the authors express CA-WASp in immature or semi-mature DC, stiffen them and record stronger T cell responses. It seems essential to be able to measure some surface parameters in these experiments – most importantly – the pMHC density. This can be accomplished with available antibodies to the SIINFEKL-Kb complex – so it might be appropriate to use OTI in this study to allow us of this reagent – which I think may be the only one for the OVA system. Since anti-CD28 costimulatory signals can be presented from solution and are not sensitive to stiffness anyway (which was published in Judokusumo et al., 2012), it should be possible to match pMHC with immature DC without and with CA-WASp and soluble anti-CD28 costim to determine if T cell respond better to stiffer immature DC generated with CA-WASp (assuming this is true). Along these lines, it was surprising that the DC didn't at all take on the stiffness of the substrate. This may be due to poor DC adhesion, particularly for mature DC. Can the author confirm that the DC can spread on the hydrogels and if not, can the authors add attachment factors or molecules like podoplanin to be able to enhance spreading and establish a link between the substrate and cortical stiffness for DC? This might be a good alternative approach if the first approach fails. Of course, the author can solve this problem in other ways, but it would much more convincing to make DC better by making them stiffer than to make them worse by removing WASp, which may also affect protrusion dynamics and change many things.

3) The result that peripheral blood cells from mouse and human are not sensitive to the substrate stiffness is interesting, but not very well documented. In several instances the authors simply show signal dye dilution profiles as the only results to make a point. For all results, the authors must provide quantification for multiple experiments (mouse) or donors (human). This also applies to results in Figure 6. When there is quantification, the authors simply use percent cell division as a method to quantify the dye dilution experiments, but this throws out a lot of information and it would be better to use a proliferation index algorithm that take into account all the data through fitting the number of cells in each division number. Hopefully this data is all available and just needs to be presented with greater rigor. The other issue is that earlier studies that showed stiffness sensitivity of human peripheral blood T cells use much stiffer PDMS elastomers so perhaps peripheral blood cells are sensitive to greater stiffnesses, perhaps related to coping with conditions in circulation. I don't feel the author need to address this last point, but they just need to acknowledge the limitations of their measurements.

[Editors’ note: the authors submitted for reconsideration following the decision after peer review. What follows is the decision letter after the second round of review.]

Thank you for submitting your work entitled "Mouse T-cell priming is enhanced by maturation-dependent stiffening of the dendritic cell cortex" for consideration by *eLife*. Your article has been reviewed by a Senior Editor, a Reviewing Editor, and two reviewers. The following individuals involved in review of your submission have agreed to reveal their identity: Claire Hivroz (Reviewer #3).

Our decision has been reached after consultation between the reviewers. Based on these discussions and the individual reviews below, we regret to inform you that your work will not be considered further for publication in *eLife*.

The reviewers and reviewing editor appreciate your ability to provide the AFM measurements on the hydrogels and the details of the indentation depth at which the DC measurements were made. These additions improve the paper. However, there are concerns about both the reliability of the commercial hydrogels and the lack of correlation between DC stiffness and T cell response in the data series manipulating WASp and other actin regulators/inhibitors. The reproducibility of the materials is a significant issue, and while it is out of your control to some extent, the possibility exists that the initial results were the outliers and the new results with the present generation of commercial gels are a better reflection the actual ability to T cells to resolve these differences in substrate stiffness. You also state that you have performed a range of manipulations to try to alter DC stiffness and have only obtained the desired correlation with WASp-KO in mature DC. This, along with the data you present in the revision, seems like a multiple testing problem that could invalidate the significance of the WASp-KO effect. But even if this is not the case, the problem still exists that details of what each manipulation does to antigen presentation by DC seems to dominate the effects of stiffness. If you can take all the efforts to manipulate DC cytoskeleton and plot stiffness against T cell response to achieve a correlation, then this would suggest an effect that cuts across the particular molecular details. But given that this doesn't seem to be the case, the WASp-KO effect in mature DC may also be an outlier that fits the model, but there are equal numbers of results that contradict the conclusion in an unbiased inspection of all the data. We greatly appreciate your sharing the information with us very openly. Unfortunately, the conclusions are not well enough supported for publication in *eLife* at this time.

Reviewer #2:

This revised manuscript substantially addresses the critiques of the initial report. The details on rigidity measurement and interpretation are important and have been addressed. I also applaud the efforts taken to understand and resolve issues regarding the Matrigen substrates. It is unfortunate that the associated experiments were not able to be carried out, but as noted by the authors those were not the main impact of this report. I support acceptance of this manuscript for publication.

Reviewer #3:

In their revised manuscript Daniel Blumenthal and co-workers have answered some of the essential revisions required by the reviewers. In particular, they really explain more thoroughly the methods used to measure the stiffness of the cells. They also present the data with more rigor.

Concerning the point raised by the reviewers on the fact that the lack of efficiency of WASp deficient dendritic cells to activate T cells can be related to something else that the viscoelastic properties, the authors have been more cautious in the text. Yet, they did not investigate if the phenotype of DC is modified at least for the main actors of Ag presentation MHC class II expression, CD28-ligands expression, ability of the cells to present peptide versus processed Ag (OVA). Thus, it is very difficult to draw any conclusion from results presented in Figure 6. are less efficient than wild type dendritic cells to activate T cells. Moreover, the overexpression of WASp in the DC, although increasing the Young modulus of T cells, decreased the ability of DC to activate T cells. This shows that the interpretation of the results on WASp-KO cells just in term of stiffness is not correct.

Another point raised by the reviewers was that it was surprising that the dendritic cells did not take on the stiffness of substrate. It is indeed very different from other adherent cells. Although the authors state in their new manuscript that DC spread and attach to the PLL coated gels it would be important to show these data and document qualitatively and quantitatively this point in the different conditions used.

One of the points was also to document better the results obtained on peripheral T cells. Yet, because it was not possible to perform new experiments with gels of different stiffness due to problem of production of the gels by the manufacturer, some of the experiments asked by the reviewers could not be performed. In their new manuscript, the authors thus took out all the experiments performed on the peripheral T cells showing that peripheral T cells and T cells from tissues do not behave the same. As a result, some of the original aspects of the previous manuscript are not present anymore.

In conclusion, in its present form, although the experiments presented in the revised manuscript are well performed, they add too little to the understanding of the modifications of the viscoelastic properties of DC and mainly present data that have already been published in other models. I thus think that the manuscript does not full fill the requirement to be published in *eLife*.

[Editors’ note: further revisions were suggested prior to acceptance, as described below.]

Thank you for submitting your article "T cell priming is enhanced by maturation-dependent stiffening of the dendritic cell cortex" for consideration by *eLife*. Your article has been reviewed by three peer reviewers, one of whom is a member of our Board of Reviewing Editors, and the evaluation has been overseen by Anna Akhmanova as the Senior Editor. The following individuals involved in review of your submission have agreed to reveal their identity: Claire Hivroz (Reviewer #1); Morgan Huse (Reviewer #3).

The reviewers have discussed the reviews with one another and the Reviewing Editor has drafted this decision to help you prepare a revised submission.

Summary:

Your study addresses the question of the mechanosensitivity of mouse T lymphocytes, in particular their sensitivity to stiffness.

You perform AFM on mouse dendritic cells (BMDC and ex vivo splenic DC) and measure the Young's modulus of the cortex of immature and LPS-matured DC and show that mature DC have an increased stiffness. You show that increasing stiffness (in the range measured for mature DC) increases the sensitivity of mouse CD4 more than of mouse CD8 T lymphocytes to low concentration of agonist peptide-MHC complexes. Exploring the role of the actin cytoskeleton in the stiffness of mature DC, they show that the Arp2/3 complex, the formins and WASp are involved. Finally, you show that DC from WASp-KO mice are less efficient than WT DC for T cell priming.

Your study is performed with care and is particularly interesting since it represents a coherent and rather complete analysis of the response of mouse T cells to stiffness: CD4 and CD8 T cells, activation with peptide-MHC complex with a careful assessment of T cell responses including measurement of CD69 and CD25, plus IL-2 secretion and cell division.

Essential revisions:

1) Interpretation of AFM-based data of Young's modulus needs to be developed more. Indentation depths appear to be in the range of several micrometers for the cells, which is on the order of cellular structures. The Hertzian model assumes a homogenous Young's modulus for several multiples of the indentation depth. Such distances include cell cytoplasm and may also reflect the rigidity of the underlying substrate. Figure 1D provides some evidence that this is not the case, but given the overall small impacts on measured modulus and comparatively soft nature of the dendritic cells, it is still possible that flattening of the cells upon maturation contributes to the changes in calculated modulus. Reanalysis of the force-indentation curves over lower ranges of indentation would help address this issue.

2) It would also be helpful for you to report on the adhesion of the DC on different rigidity substrates as DC are not typically highly adherent cells, but you are using a unique substrate so this is uncharted territory. Degree of adhesion and cell height above the substrate under all relevant conditions would be important data that could be conveyed in a table.

3) As only one TCR-pMHC combination is studies for CD4 and CD8 T cells, it’s possible that the different response might be a function of the potency of the pMHC, rather than a difference between CD4 and CD8 T cells. Do you have data comparing naive CD4 and CD8 T cells with otherwise identical anti-CD3 substrates? This might still provide the best comparison as supplemental data, although less physiological than the pMHC data.

4) You could discuss the potential impact of differences in lateral mobility of pMHC on the hydrogels vs DC.

[Editors’ note: further revisions were suggested prior to acceptance, as described below.]

Thank you for submitting your article "Mouse T cell priming is enhanced by maturation-dependent stiffening of the dendritic cell cortex" for consideration by *eLife*. Your article has been reviewed by three peer reviewers, one of whom is a member of our Board of Reviewing Editors, and the evaluation has been overseen by Anna Akhmanova as the Senior Editor. The following individuals involved in review of your submission have agreed to reveal their identity: Claire Hivroz (Reviewer #1); Morgan Huse (Reviewer #3).

The reviewers have discussed the reviews with one another and the Reviewing Editor has drafted this decision to help you prepare a revised submission.

Your carefully conducted study is particularly interesting since it represents a coherent and complete analysis of the response (expression of activation markers, IL-2 production and proliferation) of mouse T cells to stiffness. It also demonstrates that the program of maturation of DC includes changing in their mechanical properties that is most probably involved in their ability to activate T lymphocytes. Your work also open new fields of investigation for example: what are the mechanisms involved in the modification of the DC cytoskeleton leading to changes in their stiffness? What are the molecular mechanisms, which explain the difference of sensitivity to stiffness between CD4 and CD8 T cells?

However, there were still some concerns about the discussion of the AFM results. No additional experiments are suggested, the reviewers concurred that the second paragraph of the Discussion section (discussing different methods of measuring Young's modulus) would be a natural place for raising these issues. You should discuss further how cortex, nucleus, invaginations and projections at the DC surface can affect the AFM measurements. In this paragraph, the authors may also comment on the limits of the technique on cells such as DC that are known for their complex surface topology.

---

## [Author Response]

Essential revisions:1) AFM measurements – Earlier measurements from Bufi et al., 2015 showed the same trend as here, but lower values. P.-H. Wu et al., 2018. shows that AFM probes of different dimensions generate different stiffness values with smaller probes giving higher values of Young's modulus. Since Bufi et al., used an infinitely large probe (a flat paddle) and this study used a 1 µm bead as the probe that larger value measured here may be a consequence of this and this should be discussed.

We agree with the reviewer – it is well known that differences in technical approach (AFM probes, whole cell rheology, etc.) can yield significant differences in absolute Young’s Modulus values. This study has now been cited, and the basis for the difference in absolute measurements has been addressed in the Discussion section.

The gels that are used for the T cell activation experiments are commercial and their stiffness is presumably specified by the supplier. The authors should measure the stiffness of these gels under the same conditions as the cells and determine the measured stiffness, which will ensure that the gels truly replicate the cellular stiffness.

We absolutely agree and we had done this from the outset but had not presented the data. These measurements are now shown in the new Figure 1—figure supplement 1 and Figure 2—figure supplement 1.

There are also some details of the measurements that should be better described. The indentation length cannot be of 5 μm since this is more the tip size (1 μm bead)…I think this is probably the range of vertical displacement of the cantilever, the motion of the cantilever being stopped when the force it is facing reaches 6 nN. As far as I understand, this indicates a cantilever deflection of 0.1 μm (100 nm), but does not say anything about the indentation depth in the cell, and this is important information to evaluate to what extent it is possible to claim that the authors have only probed the cortex of the cells. Finally, the authors should also explain the meaning of the 2 Hz acquistion frequency. Contact mode measurements are not supposed to be sine waves. So, I think this is the frequency at which measurements were repeated. It would be better to indicate the approach/retraction rate, i.e. a speed in μm/s. This parameter has indeed an important impact on the measured stiffness value. The authors should also provide information about where on the cell they were indenting and if the location mattered?

We apologize if this was not clear. The reviewer is correct – 5μm refers to the total z-displacement of the AFM cantilever, which is stopped when the force reaches 6nN. We have expanded the explanation of AFM measurements in the Materials and methods section and have included information on the approach/retraction speed, and where in the cell measurements were made (near, but not over the nucleus). These details are also stressed in the Results section. The actual indentation length (roughly 1.5 μm) was quantified in representative experiments and is now shown in the new Figure 1—figure supplement 2.

2) The experiments to manipulate DC stiffness are important to the impact of the paper beyond earlier work. It’s interesting that mature DC lacking WASp are both softer and less potent in T cell stimulation, but there may be many other explanations for this. The disappointing result that CA-WASp didn't further increase T cell responses might reflect saturation of the response to stiffness.

We agree that the decrease in stiffness and potency of WASp-KO DCs could have other explanations, and so took care not to overinterpret this result. With respect to mature DCs expressing CA-WASp, our data using the hydrogel system clearly show that the T cell stiffness response is not saturated. However, the relatively subtle increase in stiffness that we observe in mature DCs transduced with CA-WASp (approx. 4 kPa to approx. 5 kPa) is not great enough to result in an increased T cell response. We included a figure (now Figure 6—figure supplement 1) to make this comparison clearer to the reader.

Can the authors express CA-WASp in immature or semi-mature DC, stiffen them and record stronger T cell responses. It seems essential to be able to measure some surface parameters in these experiments – most importantly – the pMHC density. This can be accomplished with available antibodies to the SIINFEKL-Kb complex – so it might be appropriate to use OTI in this study to allow us of this reagent – which I think may be the only one for the OVA system. Since anti-CD28 costimulatory signals can be presented from solution and are not sensitive to stiffness anyway (which was published in Judokusumo et al., 2012), it should be possible to match pMHC with immature DC without and with CA-WASp and soluble anti-CD28 costim to determine if T cell respond better to stiffer immature DC generated with CA-WASp (assuming this is true).

This is a very interesting and constructive suggestion. We had already tried several approaches to modifying DC stiffness. In addition to the HS1 and fascin knockouts shown in the paper, we tried FMNL1/m-Dia-1 double knockouts, transduction with constitutively active and dominant negative RhoA, and overexpression of L-plastin and T-plastin, as well as several other pharmacological strategies. None of these approaches had a significant effect on stiffness of mature DCs as measured by AFM. In response to the reviewer’s promising suggestion, we attempted to modulate the stiffness of immature DCs. We transduced BM precursors with CA-WASp or a control, selected transduced cells without lipopolysaccharide stimulation, and analyzed T cell priming using the OTII system. As shown below, T cell priming was actually *diminished by CA-WASp expression*, even when CD28 costimulation was provided in trans. AFM analysis of the transduced cells revealed that when immature DCs are made to express CA-WASp, there is no significant increase in cortical stiffness. This is likely because WASp plays a very different role in immature vs mature DCs. In immature DCs, it is primarily deployed to the ventral surface, where it is essential for the formation of podosomes. In keeping with this, we showed in Figure 2 that depolymerization of F-actin has no impact on the stiffness of immature DCs. These new data suggest that even if you force expression of active WASp in these cells, it has little or no impact on the properties of the cell cortex. We agree with the reviewer that if the forced expression of CA-WASp had, as hoped, increased stiffness and T cell priming, we would have needed to normalize for expression of pMHC and costimulatory ligands. However, under the circumstances, there was little point in pursuing those controls. Based on this analysis, we do not think that the cortical cytoskeleton in immature DCs can be manipulated in interpretable ways, even if one compensates for other features of maturation. The data on CA-WASp-transduced immature DCs is shown in Author response image 1. We opted not to incorporate it into the revised manuscript because we felt that it was not particularly illuminating for the reader. However, if the reviewers feel that it should be included, these data could be incorporated into Figure 6, or provided as an additional supplemental figure.

**Author response image 1. sa2fig1:** (A) CFSE dilution plots comparing T cell priming by WT (grayfilled histogram) or CA-WASp (redopen histogram) immature DCs in the presence of a range of OVA peptide with CD28 costimulation given in trans. (B) AFM analysis of CA-WASp transduced cells. immature CA-WASp DC stiffness (green) is shown side-by-side with older data (Figure 6 in the manuscript) to facilitate comparison with WT CA-WASp mature cells.

Along these lines, it was surprising that the DC didn't at all take on the stiffness of the substrate. This may be due to poor DC adhesion, particularly for mature DC. Can the author confirm that the DC can spread on the hydrogels and if not, can the authors add attachment factors or molecules like podoplanin to be able to enhance spreading and establish a link between the substrate and cortical stiffness for DC? This might be a good alternative approach if the first approach fails. Of course, the author can solve this problem in other ways, but it would much more convincing to make DC better by making them stiffer than to make them worse by removing WASp, which may also affect protrusion dynamics and change many things.

The reviewer is correct that attachment of mature DCs to the hydrogels (actually to any surface) is problematic. For this reason, we have always attached them using Poly L-lysine, which we found to be superior to fibronectin. Thus, all cells in the study were well adhered to the substrate. We have clarified this in the text and have specified that we observed no change in the extent of cell spreading or morphology on the different hydrogels. The non-responsiveness of DCs to substrate stiffness and their failure to undergo significant increases in stiffness after genetic manipulation suggests that these cells maintain homeostatic control of cortical stiffness, regardless of external manipulations. This property is reminiscent of their tendency to migrate at a constant rate as they move along substrates with different adhesive properties, as characterized by the Sixt lab (Renkawitz et al., 2009).

3) The result that peripheral blood cells from mouse and human are not sensitive to the substrate stiffness is interesting, but not very well documented. In several instances the authors simply show signal dye dilution profiles as the only results to make a point. For all results, the authors must provide quantification for multiple experiments (mouse) or donors (human). This also applies to results in Figure 6. When there is quantification, the authors simply use percent cell division as a method to quantify the dye dilution experiments, but this throws out a lot of information and it would be better to use a proliferation index algorithm that take into account all the data through fitting the number of cells in each division number. Hopefully this data is all available and just needs to be presented with greater rigor. The other issue is that earlier studies that showed stiffness sensitivity of human peripheral blood T cells use much stiffer PDMS elastomers so perhaps peripheral blood cells are sensitive to greater stiffnesses, perhaps related to coping with conditions in circulation. I don't feel the author need to address this last point, but they just need to acknowledge the limitations of their measurements.

We have now added quantification from multiple experiments as division index for all figures involving T cell proliferation, retaining primary data (CFSE plots) from single representative experiments as well, where this will be helpful to the reader. As detailed in the cover letter, we have eliminated studies involving peripheral blood T cells.

[Editors’ note: the authors resubmitted a revised version of the paper for consideration. What follows is the authors’ response to the second round of review.]

The reviewers and reviewing editor appreciate your ability to provide the AFM measurements on the hydrogels and the details of the indentation depth at which the DC measurements were made. These additions improve the paper. However, there are concerns about both the reliability of the commercial hydrogels and the lack of correlation between DC stiffness and T cell response in the data series manipulating WASp and other actin regulators/inhibitors. The reproducibility of the materials is a significant issue, and while it is out of your control to some extent, the possibility exists that the initial results were the outliers and the new results with the present generation of commercial gels are a better reflection the actual ability to T cells to resolve these differences in substrate stiffness.

We completely agree with the reviewers on this point. Indeed, the old set of hydrogels were found to have several issues. For that reason, we have taken every precaution with the new hydrogel surfaces. We have worked closely with the manufacturer to ensure that their new surfaces are consistent from batch-to-batch over the course of many months, that they bind stimulatory ligands in a reproducible manner, and that they have no physical defects (e.g. shrinkage exposing plastic surface). We are now confident that our results are reproducible within our own lab, and that others should be able to reproduce and build on our findings. In addition, we have taken several steps to ensure that our new results reflect the actual ability of T cells to sense substrate stiffness:

1) We have redone all the studies in the manuscript that involve T cell activation by hydrogel substrates with the new hydrogel surfaces, with a high number of repeats to ensure reproducible results.

2) To better examine T cell stiffness sensing, we have incorporated several new ways to measure T cell activation in response to stimulatory hydrogel surfaces:

a) In a set of new experiments, we looked at T cell activation at an earlier time point, by measuring upregulation of the activation markers CD69 and CD25, and IL-2 production at 24 hours post stimulation. This new time point reveals an intriguing difference between CD4^+^ and CD8^+^ T cells. While CD4^+^ T cells showed profound stiffness-dependent responses, CD8^+^ T cells were much less stiffness dependent.

b) In these new studies, we measure expression of CD25 in all CFSE dilution proliferation assays (done at 72 hours post stimulation). This new addition validates our results further and adds a new facet to understanding the phenomenon.

3) We have added a new set of experiments where we compare T cell activation by pMHC, anti-CD3ε, and anti-TCRvβ side-by-side. These studies reveal that stiffness-dependent responses are best detected when T cells are stimulated by pMHC. This new set of studies adds to the manuscript by relating T cell mechanosensing to the current models of TCR deformation.

You also state that you have performed a range of manipulations to try to alter DC stiffness and have only obtained the desired correlation with WASp-KO in mature DC. This, along with the data you present in the revision, seems like a multiple testing problem that could invalidate the significance of the WASp-KO effect. But even if this is not the case, the problem still exists that details of what each manipulation does to antigen presentation by DC seems to dominate the effects of stiffness. If you can take all the efforts to manipulate DC cytoskeleton and plot stiffness against T cell response to achieve a correlation, then this would suggest an effect that cuts across the particular molecular details. But given that this doesn't seem to be the case, the WASp-KO effect in mature DC may also be an outlier that fits the model, but there are equal numbers of results that contradict the conclusion in an unbiased inspection of all the data. We greatly appreciate your sharing the information with us very openly. Unfortunately, the conclusions are not well enough supported for publication in eLife at this time.

There may be some misunderstanding here. We never obtained any data in which the correlation between DC stiffness and T cell responses was contradictory to our findings using hydrogels. The problem was that we could not manipulate DC stiffness. We tried numerous ways to alter DC stiffness for the previous submission and have tried more since then. These include knockouts of HS1, moesin, Fascin-1, L-plastin, T-plastin Ena/Vasp, and mDia/FMNL1, expression of constitutively active and dominant negative Rho-A, overexpression of a constitutively active moesin mutant, treatment with RhoA activator, and light fixation. Some of these manipulations were not tolerated by the cells, while other did not significantly alter cortical stiffness based on AFM analysis. It seems that DCs maintain constant cortical stiffness in the face of many cytoskeletal manipulations. Thus, while we would very much like to plot DC stiffness against T cell responses, we did not even test T cell responses in most cases, as correlative results where the variable in question was not altered would not be informative. It is disappointing we were unable to manipulate DC stiffness, but at this point, it’s clear that the hydrogel data is the best way to test T cell stiffness responses. We have included the data on DCs lacking WASp or expressing the constitutively active WASp mutant, as these were the best ways we could find to alter DC stiffness. The data do fit our model, but we have taken care in the revised manuscript to address the limitations of this experiment, and we have tried not to overstate the significance of these findings.

[Editors’ note: what follows is the authors’ response to the third round of review.]

Essential revisions:1) Interpretation of AFM-based data of Young's modulus needs to be developed more. Indentation depths appear to be in the range of several micrometers for the cells, which is on the order of cellular structures. The Hertzian model assumes a homogenous Young's modulus for several multiples of the indentation depth. Such distances include cell cytoplasm and may also reflect the rigidity of the underlying substrate. Figure 1D provides some evidence that this is not the case, but given the overall small impacts on measured modulus and comparatively soft nature of the dendritic cells, it is still possible that flattening of the cells upon maturation contributes to the changes in calculated modulus. Reanalysis of the force-indentation curves over lower ranges of indentation would help address this issue.

We totally agree with the reviewer on this matter and made the following changes to address this. All AFM data was reanalyzed with new constraints on the fitting algorithm to ensure that fitting is done based on ~0.5 µm indentation depth. Since the Bruker software does not allow fitting restriction by Z axis separation, we had to take a different approach to restrict fitting to ~0.5 µm. After consultations with other scientists in the field, we achieved this by restricting the Hertzian model fitting to 30% of total force applied, which we found corresponds to ~0.5 µm indentation depth. This new procedure and the reasoning for it are now included in the Materials and methods section under the description of AFM procedures. For clarity, we have added an example of the implementation of this procedure in Figure 1—figure supplement 4. Figure 1, Figure 2 and Figure 7 were revised to reflect resulting changes to the exact stiffness values, and the accompanying text in the Results section was modified accordingly. Importantly, this new analysis did not significantly alter the overall relationships among the different samples or the interpretation of our results.

2) It would also be helpful for you to report on the adhesion of the DC on different rigidity substrates as DC are not typically highly adherent cells, but you are using a unique substrate so this is uncharted territory. Degree of adhesion and cell height above the substrate under all relevant conditions would be important data that could be conveyed in a table.

We thank the reviewer for pointing this out to us. We have used Poly-L-Lysine to stick the BMDCs down to the surfaces to minimize the effect of differences in adhesive properties of the cells, and we did not observe major differences in cell morphology during AFM measurements. To clarify this for readers, we have now added a new figure (Figure 1—figure supplement 2 ) showing the footprint (spreading area) of BMDCs (immature and mature) on hydrogel and glass surfaces (always coated with Poly L-lysine). Representative images, as well as the analysis of mean cell size are shown, and described in the manuscript. We found that all cells show a similar footprint on the different surfaces, with mature cells being slightly smaller than immature ones. Although there is one exception where immature DCs have a statistically significantly larger footprint on the 25kPa surface, the change in size is small (less than 10%).

Unfortunately, due to the covid-19 situation, we were unable to locate an upright confocal microscope to image the cells height above the surface as requested. Instead, we had to default to using a long working distance 40X objective and image the cells through the hydrogels, resulting in reduced image quality. Nevertheless, our data clearly shows that our use of PLL does, in fact, allow for similar spreading of BMDCs on the different surfaces. Together with the changes in the AFM analysis, we feel confident that cell spreading does not affect the AFM measurements, and that our results reflect a biological mechanism.

3) As only one TCR-pMHC combination is studies for CD4 and CD8 T cells, it’s possible that the different response might be a function of the potency of the pMHC, rather than a difference between CD4 and CD8 T cells. Do you have data comparing naive CD4 and CD8 T cells with otherwise identical anti-CD3 substrates? This might still provide the best comparison as supplemental data, although less physiological than the pMHC data.

The reviewer raises an excellent point, with an excellent suggestion on how to resolve this issue. We did the experiment suggested by the reviewer and the results are now given in Figure 5—figure supplement 1 and addressed in the accompanying text. Briefly, we stimulated OT-II CD4^+^ and OT-I CD8^+^ T cells side-by-side on the same set of surfaces used throughout the manuscript, using a varying concentration of αCD3ε (2C11) and constant αCD28 (PV1) T cell proliferation was measured using CFSE dilution. To facilitate comparisons between this experiment other studies using pMHC stimulation, division indices were normalized. This makes it possible to compare the trends in stiffness responses and not the actual division index numbers, which are naturally higher for antibody stimulation. We find that similar to pMHC stimulation, stimulating CD4^+^ T cells with 2C11 results in a profound stiffness dependent response, whereas soft hydrogels (2,4 kPa) do not even stimulate a response. Similarly, CD8^+^ T cells stimulated with antibody show only a weak stiffness dependency, similar to the pMHC-I stimulated counterparts. We thus conclude that the difference between CD4^+^ and CD8^+^ stiffness responses is not due to different antigen strengths but representing of a different biological mechanism.

4) You could discuss the potential impact of differences in lateral mobility of pMHC on the hydrogels vs DC.

This is an important point, which will need to be addressed experimentally in future work. We have added text addressing this point in the Discussion section.

[Editors’ note: what follows is the authors’ response to the fourth round of review.]

Your carefully conducted study is particularly interesting since it represents a coherent and complete analysis of the response (expression of activation markers, IL-2 production and proliferation) of mouse T cells to stiffness. It also demonstrates that the program of maturation of DC includes changing in their mechanical properties that is most probably involved in their ability to activate T lymphocytes. Your work also open new fields of investigation for example: what are the mechanisms involved in the modification of the DC cytoskeleton leading to changes in their stiffness? What are the molecular mechanisms, which explain the difference of sensitivity to stiffness between CD4 and CD8 T cells?However, there were still some concerns about the discussion of the AFM results. No additional experiments are suggested, the reviewers concurred that the second paragraph of the Discussion section (discussing different methods of measuring Young's modulus) would be a natural place for raising these issues. You should discuss further how cortex, nucleus, invaginations and projections at the DC surface can affect the AFM measurements. In this paragraph, the authors may also comment on the limits of the technique on cells such as DC that are known for their complex surface topology.

We are pleased to resubmit this revised version of our manuscript entitled “T cell priming is enhanced by maturation-dependent stiffening of the dendritic cell cortex” for publication in *eLife*. We have addressed all the remaining reviewers’ concerns. In particular, we added a new paragraph to the Discussion section to address the topological complexity of the DC surface, and how this may influence AFM measurements. In addition, we have corrected the missing reference and made the other recommended changes to spelling and grammar. To designate plastic surfaces and avoid confusion, Pl was changed to PL in several figures and their legends, so all figures have been re-uploaded.